# Fast turnover of genome transcription across evolutionary time exposes entire non-coding DNA to *de novo* gene emergence

**Rafik Neme\*, Diethard Tautz\***

Max-Planck Institute for Evolutionary Biology, Plön, Germany

**Abstract** Deep sequencing analyses have shown that a large fraction of genomes is transcribed, but the significance of this transcription is much debated. Here, we characterize the phylogenetic turnover of poly-adenylated transcripts in a comprehensive sampling of taxa of the mouse (genus *Mus*), spanning a phylogenetic distance of 10 Myr. Using deep RNA sequencing we find that at a given sequencing depth transcriptome coverage becomes saturated within a taxon, but keeps extending when compared between taxa, even at this very shallow phylogenetic level. Our data show a high turnover of transcriptional states between taxa and that no major transcript-free islands exist across evolutionary time. This suggests that the entire genome can be transcribed into poly-adenylated RNA when viewed at an evolutionary time scale. We conclude that any part of the non-coding genome can potentially become subject to evolutionary functionalization via *de novo* gene evolution within relatively short evolutionary time spans.

## Introduction

Genome-wide surveys have provided evidence for 'pervasive transcription', i.e., much larger portions of the genome are transcribed than would have been predicted from annotated exons (*Clark et al., 2011*; *Hangauer et al., 2013*; *Kellis et al., 2014*). Most are expected to be non-coding RNAs (lncRNAs) of which some have been shown to be functional. However, the general conservation level of these additional transcripts tends to be low, which raises the question of their evolutionary turnover dynamics (*Kutter et al., 2012*; *Kapusta and Feschotte, 2014*). They are currently receiving additional attention, since they could be a source for *de novo* gene formation via a proto-gene stage (*Carvunis et al., 2012*; *Ruiz-Orera et al., 2014*; *Neme and Tautz, 2014*). It has been shown that *de novo* gene emergence shows particularly high rates in the youngest lineages (*Tautz and Domazet-Loso, 2011*), indicating that there is high turnover of such transcripts and genes between closely related species. Indeed, comparative studies of *de novo* genes between *Drosophila* species (*Palmieri et al., 2014*) and within *Drosophila* populations (*Zhao et al., 2014*) have confirmed this.

A number of possibilities have been discussed by which new transcripts are generated in previously non-coding regions, including single mutational events, stabilization of bi-directional transcription and insertion of transposable elements with promotor activity (*Brosius, 2005*; *Gotea et al., 2013*; *Neme and Tautz, 2013*; *Wu and Sharp, 2013*; *Sundaram et al., 2014*; *Ruiz-Orera et al., 2015*). Detailed analyses of specific cases of emergence of a *de novo* gene have shown that single step mutations can be sufficient to generate a stable transcript in a region that was previously not transcribed and translated (*Heinen et al., 2009*; *Knowles and McLysaght, 2009*). The unequivocal identification of *de novo* transcript emergence can only be made in a comparison between very closely related evolutionary lineages, where orthologous genomic regions can be fully aligned, even

**\*For correspondence:** rneme@ evolbio.mpg.de (RN); tautz@ evolbio.mpg.de (DT)

**eLife digest** Traditionally, the genome – the sum total of DNA within a cell – was thought to be divided into genes and 'non-coding' regions. Genes are copied, or "transcribed", into molecules called RNA that perform essential tasks in the cell. The roles of the non-coding regions were often less clear, although it has since become apparent that some are also transcribed and generate low levels of RNA molecules. However, many debate how significant this transcription is to living organisms.

Neme and Tautz have now used a technique called deep RNA sequencing to study the RNA molecules produced in several different species and types of mice whose last common ancestor lived 10 million years ago. Different species produced RNA molecules from different portions – both genes and non-coding regions – of their genomes. Comparing these RNA sequences suggests that changes to the regions that are transcribed occur relatively quickly for a large portion of the genome. Furthermore, there have been no significant areas of the common ancestor's genome that have not been transcribed at some point in at least one of its descendent species.

This therefore suggests that over a relatively short evolutionary period, any part of the genome can acquire the ability to be transcribed and potentially form a new gene. The next challenge is to find out how often these transcribed non-coding parts of the genome show important biochemical activities, and how they find their way into becoming new genes.

for the neutrally evolving parts of the genome (*Tautz et al., 2013*). While the available genome and transcriptome data for mammals and insects are sufficient to screen for specific cases of *de novo* transcript emergence, they are still too far apart of each other to allow a comprehensive genome-wide assessment. Our analysis here is therefore based on a new dataset that reflects a very shallow divergence time-frame for relatives of the house mouse (*Mus musculus*).

## Results

We selected populations, subspecies and species with increasing phylogenetic distance to the *Mus musculus* reference sequence (*Keane et al., 2011*). This reference was derived from an inbred strain of the subspecies *Mus musculus domesticus* and we use samples from three wild type populations of *M. m. domesticus* as the most closely related taxa, separated from each other by about 3,000–10,000 years. Further, we use samples from the related subspecies *M. m. musculus* and *M. m. castaneus*, which are separated since 0.3–0.5 million years. The other samples are recognized separate species with increasing evolutionary distances (*Figure 1*). We call this set of populations, subspecies and species collectively 'taxa' in the following. Altogether they span 10 million years of divergence, which corresponds to an average of 6% nucleotide difference for the most distant comparisons.

We obtained genome sequence reads for all taxa and mapped them to the mouse reference genome, using an algorithm that was specifically designed to deal efficiently with problems that occur in cross-mapping between diverged genomes (*Sedlazeck et al., 2013*; see Appendix 1 for validation). All regions that could be unequivocally mapped for all taxa were then used for further analysis. We refer to this as the 'common genome' which allows comparisons on those regions of the genomes which have not been gained or lost along the phylogeny, i.e., are common across all taxa (*Figure 1—figure supplement 1*). It represents 71.7% of the total reference genome length (*Figure 1—figure supplement 2*). Hence, we are nominally not analyzing about a third of the total genome length, but this corresponds to the highly repetitive parts for which unique and reliable mapping of transcriptomic reads would not be possible. Also, changes in transcription derived from gain or loss of genomic regions do not contribute to the patterns described below.

We chose three tissues for transcriptome sequencing, including testis, brain and liver. Previous studies had shown that testis and brain harbor the largest diversity of transcripts (*Necsulea and Kaessmann, 2014*). We sequenced only the poly-A$^+$ fraction of the RNA, i.e., our focus is on coding and non-coding exons in processed RNA.

We use non-overlapping sliding windows of 200nt to assay for presence or absence of reads within the windows and express overall coverage as the fraction of windows showing transcription

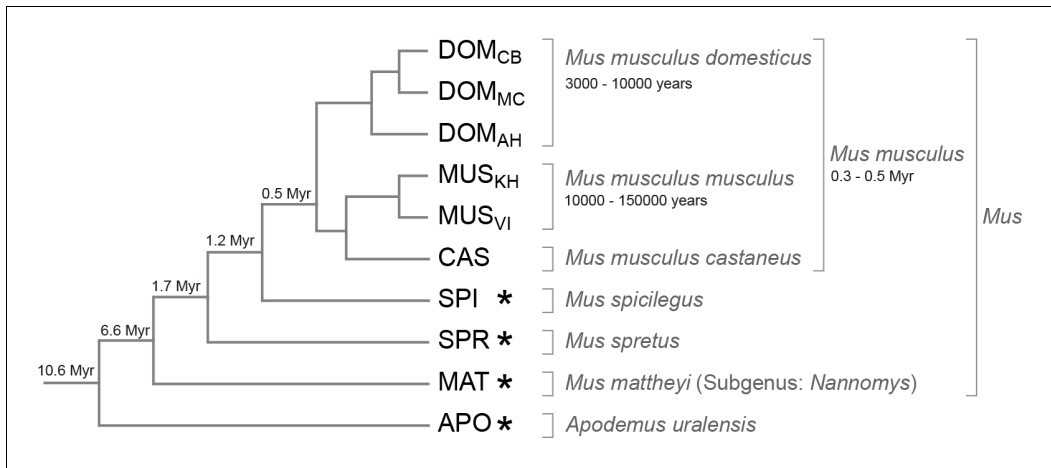

**Figure 1.** Phylogenetic relationships and time estimates for the taxa used in the study. New genome sequences were generated for taxa with *. A common genome was constructed across all taxa (*Figure 1—figure supplement 1*) based on a mapping algorithm that is not affected by the sequence divergence between the samples (Appendix 1). *Figure 1—figure supplement 2* shows the intersection of genome coverage between the named species.

The following figure supplements are available for figure 1:

**Figure supplement 1.** Scheme for the establishment of the 'common genome' using genomic reads and the mouse reference genome.

**Figure supplement 2.** Venn diagrams of representation of the common genome, derived from 200bp windows covered in genomic reads in species with more than one million years divergence to the reference.

(see methods for details). We use only uniquely mapping reads, implying that we neglect the contributions and dynamics at repetitive loci. We display three thresholds of window coverage, the minimum being coverage by at least a single read, while the higher ones represent at least 10 and 100 reads respectively. The first serves as a very inclusive metric of low-level transcription, with the drawback of potentially including noise into the analysis, due to stochasticity in sampling, while the others represent thresholds for more abundant transcripts that are unlikely to be affected by sampling noise.

Among the three tissues analyzed, liver has the lowest overall read coverage while brain and testis have similar overall levels (*Figure 2A–C*). Combining the data from all three tissues or triplicating the read depth for one tissue (brain) increases the overall coverage in a similar way (*Figure 2D,E*).

*Figure 2F* shows the total coverage across all tissues and all sequencing runs, which amounts to an average of 50.0 ± 2.5% per taxon. Hence, for each tissue, as well as in this combined set, we observe a very similar coverage in all taxa, with only a slight increase in the low expressed fraction for the most distant comparisons (see also legend *Figure 2*). This more or less stable pattern across phylogenetic time could either be due to the same regions being transcribed in all taxa, or a more or less constant rate of turnover of gain and loss of transcription between taxa.

To test these alternatives, we have asked which part of the transcribed window coverage is shared between the taxa and which is specific to the taxa. For this, we consider three classes: i) windows that are found in a single taxon only, ii) windows that are found in 2–9 taxa, i.e. more than one but not in all and iii) windows shared among all taxa (*Figure 3*; *Figure 3—figure supplement 1* shows an extended version where class ii) is separated into each individual group). However, such an analysis could potentially be subject to a sampling problem, i.e. not finding a transcript in a taxon does not necessarily imply true absence, but could also be due to failure of sampling. This would be particularly problematic for singleton reads, since the probability of falsely not detecting one in a second sample that expresses it at the same level is about 37%. However, given that we ask whether

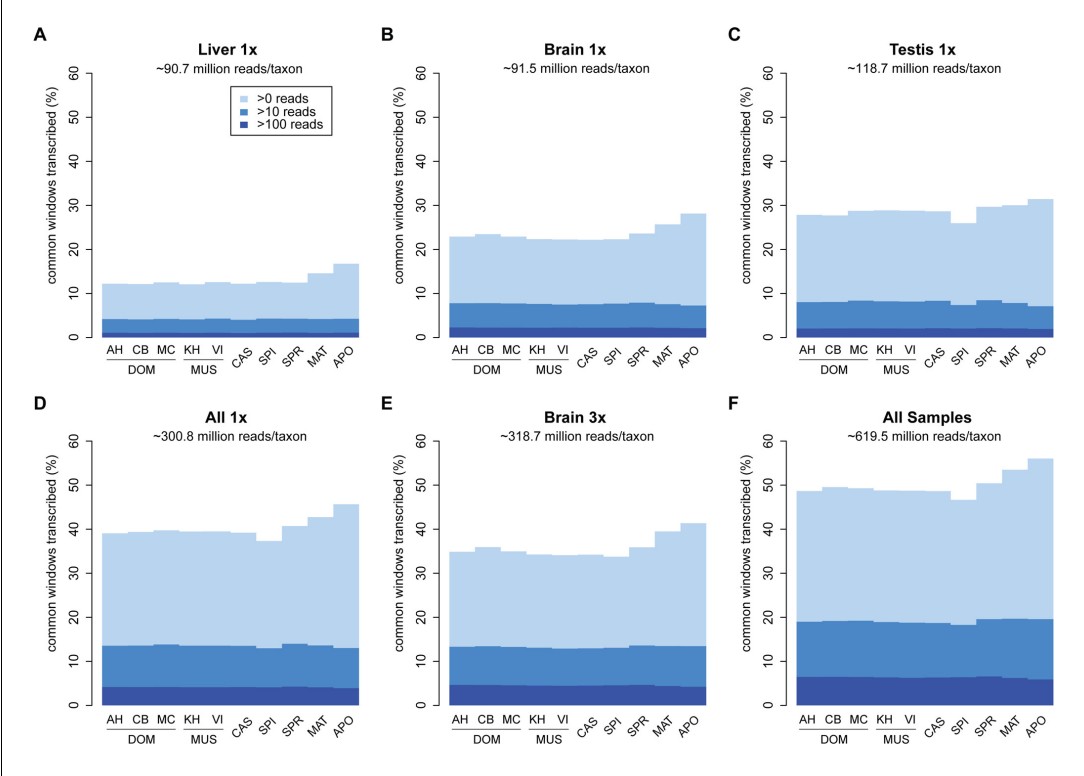

**Figure 2.** Transcriptome coverage of the common genome per taxon. (**A–C**) Liver, brain and testis, respectively, sequenced at approximately the same depth. (**D**) Combination of samples from **A–D**. (**E**) Additional sequencing of brain samples at 3x depth, compared to **B**. (**F**) Combination of all samples, including additional brain sequencing. Three coverage levels are represented by colors from light blue to dark blue: window coverage with at least 1, 10 and 100 reads. Taxon abbreviations as summarized in *Figure 1*, with closest to the reference genome to the left of each panel and most divergent one to the right. Note that the slight rise in low read coverage for the distant taxa could partially be due to slightly more mismapping of reads at this phylogenetic distance (see Appendix 1 for simulation of mapping efficiency), but is also affected by a larger fraction of singleton reads (compare *Figure 4—figure supplement 1*).

it is detected in any of the other 9 taxa, the probability of falsely not detecting it if it exists across all of them becomes small (0.01%) (see also further analysis on singletons below).

Between 1 and 7% of transcribed windows are unique to one taxon only, with the more distant taxa showing the higher percentages (*Figure 3*). Most of these taxon-specific transcripts are lowly expressed (<10 reads per window), but the more distant taxa (MAT and APO in *Figure 3I,J*) show also some more highly expressed ones. We find a total of 6566 windows with read counts >50 that occur in a single only, mostly in the long branches leading to MAT (1638 windows) and APO (4485 windows), but some also between the most closely related taxa (43 windows for DOM, including populations; 38 windows for MUS, including populations).

Approximately 18% of windows show transcripts shared across all taxa. These include most of the very highly expressed ones (>100 reads per window), but also a fraction of the low expressed ones (*Figure 3*). They are also enriched in annotated genes, especially in exons of protein coding genes, but also in non-coding genes (*Figure 3—figure supplement 2*). The class ii) windows (sharing between 2 and 9 taxa in *Figure 3*) represents the genes showing more or less turnover between taxa, with more turnover the more distant they are of each other (*Figure 3—figure supplement 1*). This class constitutes cumulatively the largest fraction (between 26 and 33% of whole genome coverage - *Figure 3*), supporting the notion of a fast turnover of most of the transcribed regions between taxa.

The taxon-specific turnover of transcripts is also reflected in a distance tree of shared coverage. Taxa that are phylogenetically closer to each other share more transcripts, i.e. the tree topology mimics that of a phylogenetic tree based on molecular sequence divergence (*Figure 4A,B*). This implies that the turnover of the transcripts is not random, but time dependent. However, the relative

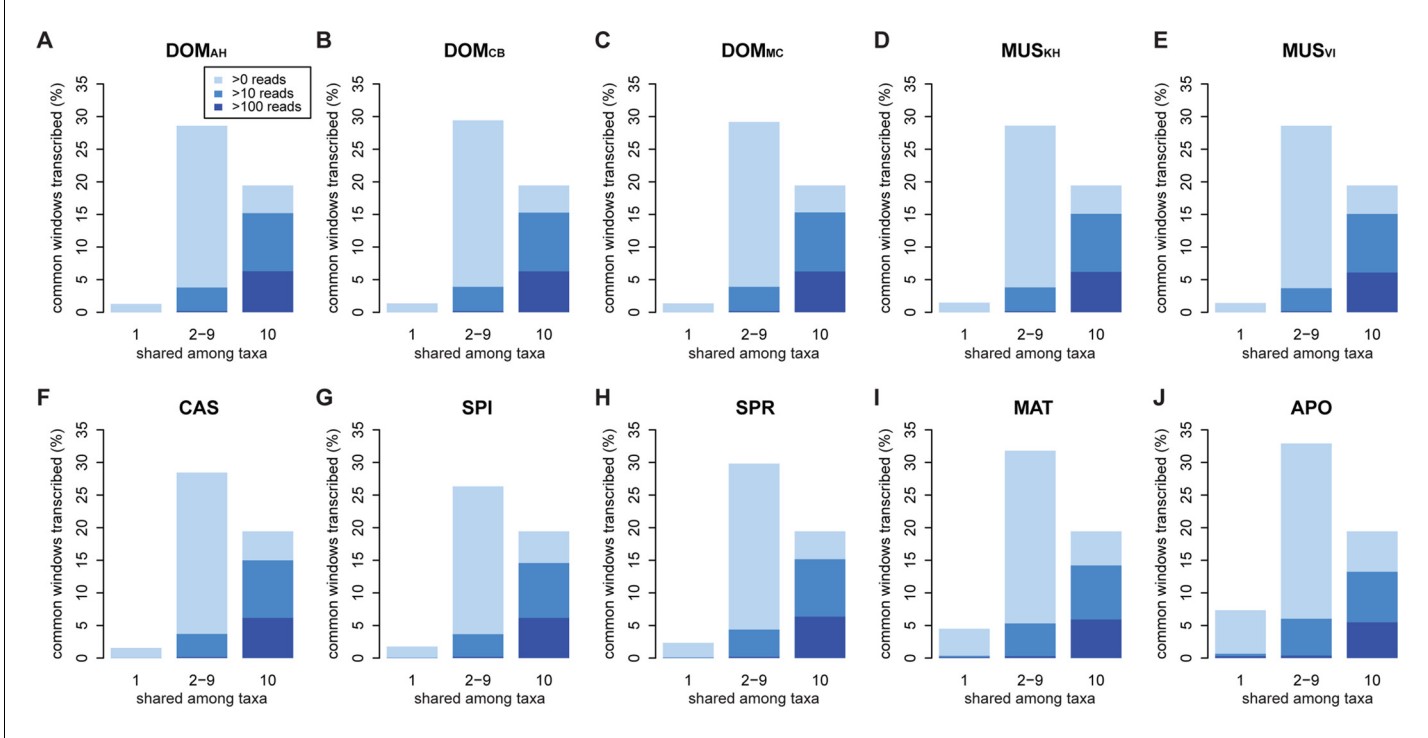

**Figure 3.** Distribution of shared and non-shared windows with transcripts for each taxon, based on the aggregate dataset across all three tissues. Three classes are represented: i) windows that are found in a single taxon only, ii) windows found in 2–9 taxa and iii) windows shared among all 10 taxa (from left to right in each panel). Windows with transcripts were first classified as belonging to one of the three classes, independent of their coverage, and were then assigned to the coverage classes represented by the blue shading (from light blue to dark blue: window coverage with at least 1, 10 and 100 reads). Taxon names as summarized in *Figure 1*. *Figure 3—figure supplement 1* shows an extended version where class ii) is separated into each individual group. Relative enrichment of annotated genes in the conserved class is shown in *Figure 3—figure supplement 2*.

The following figure supplements are available for figure 3:

**Figure supplement 1.** Distribution of shared transcripts according to the number of taxa shared, based on the aggregate dataset across all three tissues.

**Figure supplement 2.** Windows transcribed across most species (9 or more) are strongly enriched in genes known from the reference genome, while windows transcribed in some taxa (8 or less) are strongly depleted from known genes.

branch lengths are much extended for the more closely related taxa compared to the molecular distances, implying that there is a particularly high turnover between them.

To assess in how much this could be due a sampling variance problem at low expression levels, we have separately analyzed the transcripts that are represented by single reads only, since these should be most sensitive towards sampling problems. Depending on read depth and tissue, they constitute about 2–12% of the common windows when assessed on a per sample basis (*Figure 4—figure supplement 1*). However, most of these singletons in a given sample were re-detected in another tissue or another taxon (*Figure 4—figure supplement 1*), such that less than 2% are present in a given taxon (*Figure 4—figure supplement 1*) and less than 7% cumulatively throughout the whole dataset (*Figure 4—figure supplement 2*). We used the extended brain sample reads, split them into three non-overlapping sets of about 100 Mill reads for each taxon and constructed trees out of these sets using only the singleton reads. This is the equivalent of repeating the same experiment three times. We find indeed differences in the resulting trees, i.e. there is a measurable sampling variance. By constructing a consensus tree, we can partition the data into a variable and a common component. We find that 88% of the branch length is influenced by sampling variance, while the remaining 12% still recover the expected topology (*Figure 4C*). When we use a read coverage of 1–5 for the same analysis, we find that 52% of the branch length are subject to sampling

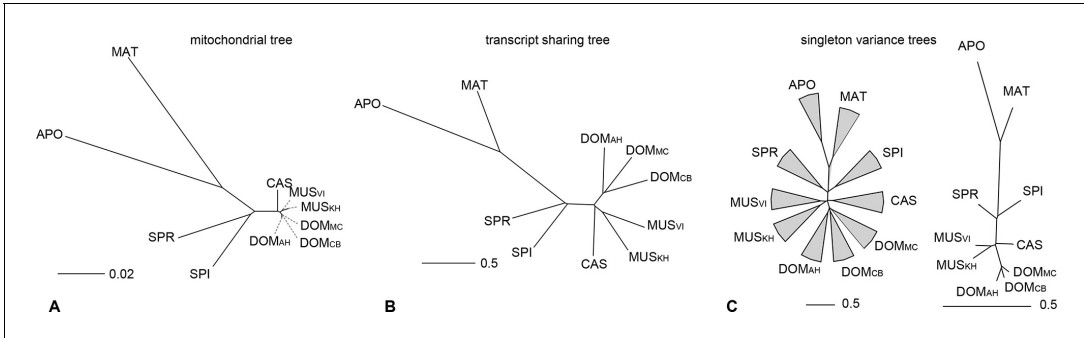

**Figure 4.** Distance tree comparisons based on molecular and transcriptome sharing data. (**A**) Molecular phylogeny based on whole mitochondrial genome sequences as a measure of molecular divergence (black lines represent the branch lengths, dashed lines serve to highlight short branches). (**B**) Tree based on shared transcriptome coverage of the genome, using correlations of presence and absence of transcription of the common genome. All nodes have bootstrap support values of 70% or more (n = 1000). (**C**) Tree based on shared transcriptome coverage of singleton reads only from subsampling of the extended brain transcriptomes. Left is the consensus tree with the variance component between samples depicted as triangles, right is the same tree, but only for the branch fraction that is robust to sampling variance. Taxon names as summarized in *Figure 1*. *Figure 4—figure supplement 1* shows the fraction of singletons in dependence of each sample in each taxon, *Figure 4—figure supplement 2* in dependence of read depth. *Figure 4—figure supplement 3* shows an extended version of the analysis shown in 4C for higher coverage levels.

The following figure supplements are available for figure 4:

**Figure supplement 1.** Fraction of windows with singletons (one paired read) of the common genome per taxon.

**Figure supplement 2.** Reduction of singletons in dependence of aggregate sequencing depth.

**Figure supplement 3.** Trees based on shared transcriptome coverage of the genome, using binary correlations.

variance and for all reads combined it is 35% (*Figure 4—figure supplement 3*). Hence, at the 100 Mill read level, we have a noticeable effect of sampling variance, but this does not erase the underlying signal. Also, the analysis in *Figure 4B* is based on 600 Mill reads per taxon, where sampling variance is expected to be further lowered.

The high dynamics of transcriptional turnover between taxa raises the question whether all parts of the genome might be accessible to transcription at some point in evolutionary time. To explore this possibility, we used a rarefaction approach to simulate the addition of one taxon at a time and used the curve to predict the behavior of adding more taxa than the ones in the present study. We compared this approach to a curve of increasing depth of sequencing, by taking subsets at 10% intervals to understand whether depth or taxonomic diversity have different behavior in this respect. We assume that in each species only a subset of the genome is transcribed, therefore the increase in depth of sequencing would saturate at some point below 100%. Conversely, if each taxon is transcribing slightly different portions of the genome due to a steady turnover, increasing the total number of sampled taxa should increase the saturation more than the increase that could be achieved by sequencing depth. This is indeed what we find. The addition of taxa indeed leads to a further increase in transcriptomic coverage, with a generalized linear model best describing the data as increasing in a logarithmic fashion (*Figure 5A*). In contrast, we observe an asymptotic behavior of the curve for increasing depth of sequencing, with apparent saturation reached at 84.1%, close to the 83.2% that we have already achieved (*Figure 5B*).

Combined with the previous results, this allows two major conclusions. First, random transcriptional noise (technical or biological) or deficiencies in sampling low level transcripts should not be major factors in our analysis, since saturation with sequencing depth would not be possible under a singleton dominated regime. Furthermore, low level transcripts (including singletons) have detectable biological signal (*Figure 4C*). Second, the data are consistent with the above outlined ideas that the evolutionary turnover leads to steady – and almost unlimited – transcriptional exploration of the genome, when summed over multiple parallel evolutionary lineages and taxa.

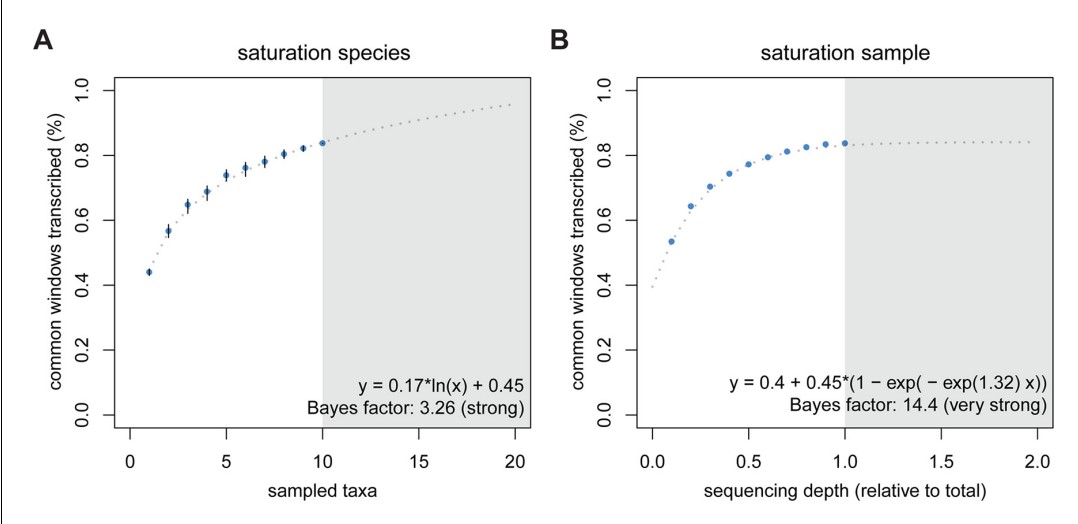

**Figure 5.** Rarefaction, subsampling and saturation patterns using all available samples and reads. (**A**) Sequencing depth saturation as estimated from an increase in the number of taxa. (**B**) Sequencing depth saturation as estimated from increasing read number. Blue dots indicate increases per sub-sampled sequence fraction or taxon added from our dataset. Gray dotted line indicates the predicted behavior from the indicated regression, and gray area shows the prediction after doubling the current sampling either by additional taxa (**A**) or in sequencing effort (**B**). Each analysis was tested for logarithmic and asymptotic models. Best fit was selected from ΔBIC, with Bayes factor shown and qualitative degree of support shown. Standard deviations are shown as black lines in **A**, and are too small to display in B (note that due to the sampling scheme for this analysis, the values above 50% are not statistically independent and that the 100% value constitutes a single data point without variance measure).

The above overall statistical consideration would still allow for the possibility of the existence of a few scattered genomic islands that are not accessible to transcription because of structural reasons (so-called transcriptional deserts – *Montavon and Duboule, 2012*) or heterochromatically packed because they are not encoding genes required in the respective tissues. Hence, we analyzed also the size distribution of transcript-free genomic regions in our dataset. We find that the maximum observed length of non-transcribed regions is 6 kb (*Figure 6*), suggesting that apparent transcriptional deserts in one taxon are readily accessible to transcription in other taxa, at least for the non-repetitive windows of the genome that are analyzed here.

## Discussion

Various studies have shown that many more regions of the genome are transcribed than are annotated as exons (*Ponting and Belgard, 2010*; *Kapranov and St. Laurent, 2012*). The significance of this additional transcription has been largely unclear and it has even been considered as noise, either biological or technical. Here we were able to trace the turnover of these extra transcripts. Our data suggest that many have sufficient stability to reflect a phylogenetic distance distribution that mimics the phylogeny of the taxa. Hence, they should not simply be considered as noise. Rather, their lifetime should be sufficient to expose them to evolutionary testing and in this way they become a substrate for *de novo* evolution of genes. On the other hand, they appear to have only a limited lifetime in case they do not acquire a function, i.e. there is also high turnover of the transcribed regions between taxa. This turnover has as a consequence that within a timespan of a few million years practically the whole genome is covered by transcription at some point in time, i.e. no major transcript-free islands exist.

We have here sampled only three tissues. If more tissues and more life stages were sampled, one would expect an even higher coverage of the genome within a given taxon. Such deep analyses have been done in the ENCODE projects (http://www.genome.gov/10005107) and they have confirmed pervasive transcription (*Clark et al., 2011*; *Hangauer et al., 2013*; *Kellis et al., 2014*) at the single-taxon level. Still, we expect that the turnover of transcribed regions between taxa would also apply to the other tissues and stages, i.e. evolutionary testing of new transcripts would relate to all

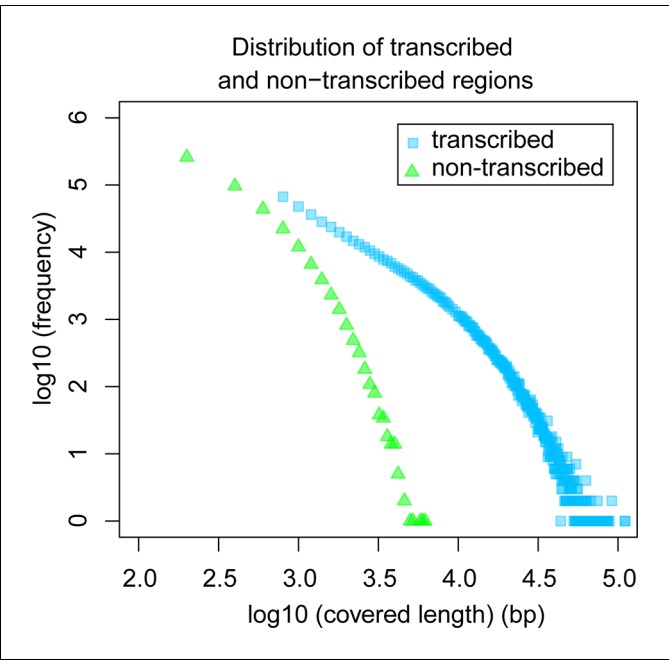

**Figure 6.** Comparative analysis of lengths of regions transcribed or not transcribed across all data (including deeper brain sequencing) in all samples. Size distribution of regions not covered in any transcript (green) versus size distribution of regions with at least one transcript (blue).

tissues and stages. This turnover is contrasted by the set of conserved genes across taxa, for which even the expression levels may be maintained across larger evolutionary distances (*Pervouchine et al., 2015*).

We see a particularly large number of lineage-specific transcripts among the most closely related taxa. This becomes most evident in the distance tree in *Figure 4B* where the branch length of the three populations of *M. m. domesticus*, which have separated only a few thousand years ago, are almost as long as those of the sister species *M. spretus* that has separated almost 2 Mill. years ago. Although this is partially influenced by sampling variance of low expressed transcripts (*Figure 4C*), this suggests that at the very short evolutionary distances (thousands of years) there is an even higher turnover of transcripts than at the longer time frames (millions of years). Such a pattern of unequal rates suggests that weak selection could act against many newly arising transcripts, such that they can exist for a short time at a population scale, but not over an extended time. Hence, we expect that the presence of such transcripts will be polymorphic at the population level, similar as it has been shown in *Drosophila* (*Zhao et al., 2014*). We have done a preliminary analysis of transcriptional variance between four individuals of each of the taxa and find this expectation fulfilled, but a more extensive study is required to obtain reliable data at this level.

We expect that a fraction of new transcripts interacts with other genes and cellular processes, either via providing a positive function or via being slightly deleterious. Our data do not allow at present to speculate on how large this 'functional' fraction would be, but this could become subject to future experimental studies. It is also as yet open whether the transcripts exert their functions as RNAs or via translation products. The analysis of ribosome profiling data has shown that many RNAs that were initially classified as non-coding can be associated to ribosomes, i.e. are likely translated (*Wilson and Masel, 2011*; *Carvunis et al., 2012*; *Ruiz-Orera et al., 2014*). On the other hand, when tracing the origin of *de novo* genes, one finds frequently that they act first as RNA and acquire open reading frames only at a later stage (*Cai et al., 2008*; *Kapranov and St. Laurent, 2012*; *Reinhardt et al., 2013* - see discussion in *Schlötterer, 2015*). For some of the *de novo* evolved genes in *Drosophila* it has been shown that they have assumed essential functions for the organism, such that knockouts of them are lethal (*Chen et al., 2010*). Global analyses of new gene emergence trends suggest that the *de novo* evolution process has been active throughout the evolutionary

history (**Neme and Tautz, 2013**). Hence, the possibility of a transition from new transcript emergence over acquisition of reading frames towards assuming essential genetic functions is well documented.

The idea that many *de novo* transcripts are slightly deleterious is concordant with the fact that various cellular processes maintain a balance between RNA transcription and degradation (**Houseley and Tollervey, 2009**; **Jensen et al., 2013**). In yeast and mammals it has been shown that several molecular pathways exist that degrade excess transcripts, in particular the ones from bidirectional promoter activity (**Jensen et al., 2013**; **Wu and Sharp, 2013**). Hence, the fact that many of the transcripts found by deep sequencing occur only at low levels does not necessarily imply a low level of transcription, but could alternatively be due to fast targeting by a degradation machinery.

Our results provide an evolutionary dynamics perspective where emergence, functionalization and decay of gene functions should be seen as an evolutionary life cycle of genes (**Neme and Tautz, 2014**). *De novo* gene birth should no longer be considered as the result of unlikely circumstances, but rather as an inherent property of the transcriptional apparatus and thus a mechanism for testing and revealing hidden adaptive potential in genomes (**Brosius, 2005**; **Masel and Siegal, 2009**). Within this evolutionary perspective, any non-genic part of the genome has the possibility to become useful at some time.

## Material and methods

### Sampled taxa

The youngest divergence point sampled, at about 3,000 years, corresponds to the split between two European populations of *Mus musculus domesticus*(**Cucchi et al., 2005**) one from France (Massif Central = $DOM_{MC}$) and one from Germany (Cologne-Bonn area = $DOM_{CB}$) (**Ihle et al., 2006**). These European populations in turn have diverged from an ancestral *M. m. domesticus* population in Iran (Ahvaz = $DOM_{AH}$) about 12,000 years ago (**Hardouin et al., 2015**). The European *M. m. domesticus* are also the closest relatives of the reference genome, the C57BL/6J strain **Didion and de Villena, 2013**).

We included two populations of *Mus musculus musculus*; one from Austria (Vienna = $MUS_{VI}$) and one from Kazakhstan (Almaty = $MUS_{KH}$). These two populations are supposed to have a longer divergence between then the European *M. m. domesticus* populations, but a more accurate estimate is currently not available. We set the divergence for analyses at around 10,000 years as an approximate estimate. *M. m. domesticus* has diverged from *M. m. musculus* and *Mus musculus castaneus* about 0.4 to 0.5 million years ago, with a subsequent divergence, not long after, between *M. m. musculus* and *M. m. castaneus* (**Suzuki et al., 2013**). We included *M. m. castaneus* (CAS) from Taiwan as a representative of the subspecies.

To account for longer divergence times, we included *Mus spicilegus* (SPI; estimated divergence of 1.2 million years); *Mus spretus* (SPR; estimated divergence of 1.7 million years)(**Suzuki et al., 2013**); *Mus matteyii* (MAT; subgenus *Nannomys*), the North African miniature mouse (estimated divergence of 6.6 million years) (**Catzeflis and Denys, 1992**; **Lecompte et al., 2008**), and *Apodemus uralensis*, the Ural field mouse (APO; estimated divergence of 10.6 million years) (**Lecompte et al., 2008**).

The population-level samples (*M. m. domesticus* and *M. m. musculus*) included are maintained under outbreeding schemes, which allows for natural polymorphisms to be present in the samples. All other non-population samples are kept as more or less inbred stock, and therefore fewer polymorphisms are expected. All mice were obtained from the mouse collection at the Max Planck Institute for Evolutionary Biology, following standard rearing techniques which ensure a homogeneous environment for all animals. Mice were maintained and handled in accordance to FELASA guidelines and German animal welfare law (Tierschutzgesetz § 11, permit from Veterinäramt Kreis Plön: 1401–144/PLÖ-004697).

A total of 60 mice were sampled, as follows: Eight male individuals from each population-level sample (outbreds), Iran ($DOM_{AH}$), France ($DOM_{MC}$), and Germany ($DOM_{CB}$) of *Mus musculus domesticus*, and Austria ($MUS_{VI}$) and Kazakhstan ($MUS_{KH}$) of *Mus musculus musculus*. Four male individuals from the remaining taxa (partially inbred): *Mus musculus castaneus* (CAS), *Mus spretus* (SPR), *Mus spicilegus* (SPI), *Mus mattheyi* (MAT) and *Apodemus uralensis* (APO). Mice were sacrificed by $CO_2$

asphyxiation followed immediately by cervical dislocation. Mice were dissected and tissues were snap-frozen within 5 min post-mortem. The tissues collected were liver (ventral view: front right lobe), both testis and whole brain including brain stem.

## Genome sequencing

One individual from each of *M. spicilegus*, *M. spretus*, *M. mattheyi*, and *Apodemus uralensis* were selected for genome sequencing. DNA was extracted from liver samples. DNA extraction was performed using a standard salt extraction protocol. Tagged libraries were prepared using the Genomic DNA Sample preparation kit from Illumina, following the manufacturers' instructions. After library preparation, the samples were run in IlluminaHiSeq 2000 at a depth of approximately 2.6 lanes per genome. Library insert size is ~190bases and paired-end reads were 100 bases long. Library preparation and sequencing was performed at the Cologne Center for Genomics. Sequencing read statistics are provided in *Table 1*. Data are available under the study accessions PRJEB11513, PRJEB11533 and PRJEB11535, from the European Nucleotide Archive (http://www.ebi.ac.uk/ena/).

## Transcriptome sequencing

The sampled tissues of each taxon were used for RNA extraction with the RNAeasy Mini Kit (QIA-GEN) and RNA was pooled at equimolar concentrations. RNA quality was measured with the Agilent RNA Nano Kit, for the individual samples and pools. Samples with RIN values above 7.5 were used for sequencing. Library preparation was done using the Illumina TruSeq library preparation, with mRNA purification (poly-A$^+$ selection), following manufacturers' instructions. Sequencing was done in Illumina HiSeq, 2000 sequencer. Libraries for each group were tagged, pooled and sequenced in a single lane, corresponding to approximately one third of a HiSeq2000 lane. Library insert size is ~190bases and paired-end reads were 100 bases long. Additional sequencing of the brain samples was performed to identify potential limitations in depth of sequencing. For this, each brain library was sequenced on a full lane of a HiSeq2000. All library preparation and sequencing was done at the Cologne Center for Genomics (CCG). Sequencing read statistics are provided in *Tables 2* and *3*. Data are available under the study accessions PRJEB11513 and PRJEB11533, from the European Nucleotide Archive (http://www.ebi.ac.uk/ena).

## Raw data processing

All raw data files were trimmed for adaptors and quality using Trimmomatic (*Lohse et al., 2012*). The quality trimming was performed basewise, removing bases below quality score of 20 (Q20), and keeping reads whose average quality was of at least Q30. Reads whose trimmed length was shorter than 60 bases were excluded from further analyses, and pairs missing one member because of poor quality were also removed from any further analyses.

## Mapping

The reconstruction of transcriptomes using high-throughput sequencing data is not trivial when comparing information across different species to a single reference genome. This is due to the fact that

**Table 1.** Genome sequencing and read mapping information relative to the C57Bl/6 reference strain (GRCm38.3/mm10).

| Species | Uniquely mapping reads (MAPQ >25) | Mean coverage depth (window based) | Reference coverage (% windows) | Total sequence divergence* | Accession Reads | Accession BAMs |
|---|---|---|---|---|---|---|
| *Apodemus uralensis* | 4.46E+08 | 40x | 78.23% | 5.60% | ERS942341 | ERS946059 |
| *Mus mattheyi* | 5.58E+08 | 52x | 77.19% | 4.50% | ERS942343 | ERS946060 |
| *Mus spretus* | 7.71E+08 | 52x | 93.91% | 1.70% | | ERS946096** |
| *Mus spicilegus* | 6.16E+08 | 57x | 84.39% | 1.60% | ERS942342 | ERS946061 |

\* The percentage of divergence was estimated from mappings using NextGenMap (*Sedlazeck et al., 2013*). Only uniquely mapping reads were considered and mapping quality greater than 25. Variation was estimated from the alignments using samtools mpileup (*Li et al., 2009*). Divergence was calculated as number of changes divided by the genome size.

\*\* Corresponds to study accession PRJEB11535. All other accessions deposited under studies PRJEB11513 and PRJEB11533.

most of the tools designed for such tasks do not work in a phylogenetically aware context. For this reason, any approximation which deals with fractional data (i.e. any high-throughput sequencing setup available to this date) is limited by the detection abilities of the software of choice and by the quality of the reference (transcriptome and genome).

Given the high quality state of the mouse genome repositories, we decided to take a reference-based approach, in which all analyses are centered in the reference genome of the C57BL/6 laboratory strain of *Mus musculus domesticus*, which enables direct comparisons across all species based on the annotations of the C57BL/6 laboratory strain.

Transcriptome and genome sequencing reads were aligned against the mm10 version of the mouse reference genome (*Waterston et al., 2002*) from UCSC (*Karolchik et al., 2014*) using Next-GenMap which performs extremely well with divergences of over 10% compared to other standard mapping software (*Sedlazeck et al., 2013*), as confirmed by our own simulations (Appendix 1). The program was run under default settings, except for –strata 1 and –silent-clip. The first option enforces uniquely mapping reads and the second drops the unmapped portion of the reads, to avoid inflating coverage statistics. This is particularly relevant around exon-intron boundaries, where exonic reads are forced into intronic regions unless this option is set.

We produced normalized versions of the alignments per tissue. This was achieved by counting the total amount of uniquely mapped reads in each taxon for each tissue, and sampling without replacement a fraction of each file which would result in the roughly the same absolute number of uniquely mapped reads for all samples of the same tissue (summarized in *Table 2* and *Table 3*).

## Coverage statistics

We performed coverage statistics on 200 bp windows, to minimize problems derived from the fractional nature of the data, in which a few nucleotides could be absent from a sequenced fragment due to the preparation of the samples, low quality towards read ends, or a few mismatches during mapping. Coverage statistics were computed from normalized alignment files with the feature-Counts program from the Subreads suite (*Liao et al., 2014*). In order to avoid counting reads twice if they would span two windows (which would be the case for most reads), we assigned reads to the window where more than half of the read was present.

Genomic reads were used as a metric of empiric mapability for the coverage statistics, i.e. to identify which regions can be reliably detected. For this, we removed from the mapping results against the reference genome (see above) all regions that were not mapped across the phylogeny based on the genomic reads from the taxa more than 1 Mill years apart. The remaining portion we call the 'common genome' in all analyses. It is important to highlight that this is not the same as synteny, since we did not perform any co-linearity analyses between fragments, but rather represent the mere presence in the species, in any possible order. The common genome serves to limit mapping artifacts, since the reads observed in each window must not only be uniquely mapping, but also be present and detectable in all the genomes considered.

We report coverage only from windows in the common genome for several reasons. First, we want to compare changes in transcription in regions which are present across all taxa, so the region must be present at the genome level. Second, the observation of transcriptome coverage on a region of the reference genome without genomic coverage from the respective taxon could represent mapping artifacts. Thus by enforcing coverage on both levels, and in all taxa at the genomic level, we reduce mapping artifacts and errors. Third, we assume that the transcriptional properties of the common genome should be general enough that they represent the properties of each of the genomes of the taxa under study. Summary data for coverage of all genomes and transcriptomes are available under the Dryad accession associated with this manuscript (doi:10.5061/dryad.8jb83).

## Reconstruction of phylogenetic relationships

We performed genome-wide correlations of coverage to infer distance between the taxa under study. Correlations of two types were initially used: rank-based (spearman correlation) and binary (phi correlation). From correlation matrices, we constructed Manhattan distance matrices and from those we further constructed neighbor-joining trees to describe the proximity between any two taxa based on shared transcriptome information. We focus mostly on the presence or absence of transcriptional coverage. For this reason, we used only the binary correlations in the figures. In this

**Table 2.** Transcriptome reads from each sample sequenced, mapped and normalized.

| Taxon Code | Tissue | Lanes | QC-passed reads | Mapped reads | (% total) | Normalized subset | (%total) | (% mapped) | Accession Reads* | Accession BAMs** |
|---|---|---|---|---|---|---|---|---|---|---|
| DOM$_{CB}$ | Brain | 0.33x | 1.30E+08 | 1.26E+08 | 96% | 9.15E+07 | 70% | 73% | ERS946023 | ERS942305 |
| DOM$_{CB}$ | Liver | 0.33x | 1.41E+08 | 1.17E+08 | 83% | 9.07E+07 | 64% | 77% | ERS946025 | ERS942306 |
| DOM$_{CB}$ | Testis | 0.33x | 1.26E+08 | 1.22E+08 | 96% | 1.19E+08 | 94% | 98% | ERS946026 | ERS942307 |
| DOM$_{MC}$ | Brain | 0.33x | 1.17E+08 | 1.13E+08 | 96% | 9.15E+07 | 78% | 81% | ERS946027 | ERS942309 |
| DOM$_{MC}$ | Liver | 0.33x | 1.34E+08 | 1.09E+08 | 81% | 9.07E+07 | 68% | 84% | ERS946029 | ERS942310 |
| DOM$_{MC}$ | Testis | 0.33x | 1.42E+08 | 1.37E+08 | 96% | 1.19E+08 | 83% | 87% | ERS946030 | ERS942311 |
| DOM$_{AH}$ | Brain | 0.33x | 9.49E+07 | 9.15E+07 | 96% | 9.15E+07 | 96% | 100% | ERS946019 | ERS942301 |
| DOM$_{AH}$ | Liver | 0.33x | 1.16E+08 | 1.02E+08 | 88% | 9.07E+07 | 78% | 89% | ERS946021 | ERS942302 |
| DOM$_{AH}$ | Testis | 0.33x | 1.61E+08 | 1.55E+08 | 96% | 1.19E+08 | 74% | 77% | ERS946022 | ERS942303 |
| MUS$_{KH}$ | Brain | 0.33x | 1.33E+08 | 1.28E+08 | 96% | 9.15E+07 | 69% | 72% | ERS946035 | ERS942313 |
| MUS$_{KH}$ | Liver | 0.33x | 1.03E+08 | 9.07E+07 | 88% | 9.07E+07 | 88% | 100% | ERS946037 | ERS942314 |
| MUS$_{KH}$ | Testis | 0.33x | 1.36E+08 | 1.31E+08 | 96% | 1.19E+08 | 87% | 91% | ERS946038 | ERS942315 |
| MUS$_{VI}$ | Brain | 0.33x | 1.23E+08 | 1.19E+08 | 96% | 9.15E+07 | 74% | 77% | ERS946031 | ERS942317 |
| MUS$_{VI}$ | Liver | 0.33x | 1.23E+08 | 9.47E+07 | 77% | 9.07E+07 | 74% | 96% | ERS946033 | ERS942318 |
| MUS$_{VI}$ | Testis | 0.33x | 1.32E+08 | 1.27E+08 | 96% | 1.19E+08 | 90% | 93% | ERS946034 | ERS942319 |
| CAS | Brain | 0.33x | 1.21E+08 | 1.16E+08 | 96% | 9.15E+07 | 76% | 79% | ERS946039 | ERS942321 |
| CAS | Liver | 0.33x | 1.23E+08 | 1.01E+08 | 82% | 9.07E+07 | 74% | 90% | ERS946041 | ERS942322 |
| CAS | Testis | 0.33x | 1.23E+08 | 1.19E+08 | 96% | 1.19E+08 | 96% | 100% | ERS946042 | ERS942323 |
| SPI | Brain | 0.33x | 1.34E+08 | 1.29E+08 | 96% | 9.15E+07 | 68% | 71% | ERS946043 | ERS942325 |
| SPI | Liver | 0.33x | 1.05E+08 | 9.82E+07 | 93% | 9.07E+07 | 86% | 92% | ERS946045 | ERS942326 |
| SPI | Testis | 0.33x | 1.44E+08 | 1.38E+08 | 96% | 1.19E+08 | 83% | 86% | ERS946046 | ERS942327 |
| SPR | Brain | 0.33x | 1.09E+08 | 1.05E+08 | 96% | 9.15E+07 | 84% | 87% | ERS946047 | ERS942329 |
| SPR | Liver | 0.33x | 1.35E+08 | 1.20E+08 | 89% | 9.07E+07 | 67% | 76% | ERS946049 | ERS942330 |
| SPR | Testis | 0.33x | 1.34E+08 | 1.29E+08 | 96% | 1.19E+08 | 88% | 92% | ERS946050 | ERS942331 |
| MAT | Brain | 0.33x | 1.12E+08 | 1.04E+08 | 93% | 9.15E+07 | 82% | 88% | ERS946051 | ERS942333 |
| MAT | Liver | 0.33x | 1.23E+08 | 1.12E+08 | 91% | 9.07E+07 | 74% | 81% | ERS946053 | ERS942334 |
| MAT | Testis | 0.33x | 1.32E+08 | 1.23E+08 | 93% | 1.19E+08 | 90% | 97% | ERS946054 | ERS942335 |
| APO | Brain | 0.33x | 1.36E+08 | 1.18E+08 | 87% | 9.15E+07 | 67% | 78% | ERS946055 | ERS942337 |
| APO | Liver | 0.33x | 1.13E+08 | 1.00E+08 | 89% | 9.07E+07 | 80% | 91% | ERS946057 | ERS942338 |
| APO | Testis | 0.33x | 1.38E+08 | 1.20E+08 | 87% | 1.19E+08 | 86% | 99% | ERS946058 | ERS942339 |

All accessions deposited under studies PRJEB11533* and PRJEB11513**.

representation, closely related organisms have more shared transcriptomic coverage than distantly related organisms. Analyses were performed in R, using the function dist() from the stats package and nj() from the ape package (*Paradis et al., 2004*).

Additionally, whole mitochondrial genomes were obtained for each taxon as consensus sequences from mapped reads using samtools mpileup (*Li et al., 2009*). The sequences were aligned with MUSCLE (*Edgar, 2004*), and a NJ tree was constructed with the dist.dna() and nj() functions from the ape package *Paradis et al., 2004*). All trees were tested with 1000 bootstraps with the boot. phylo() function from the ape package. Reported nodes have a support of 70% or greater.

## Estimation of sampling variance from brain samples

The extensive sequencing of brain samples were used to obtain estimates of how sampling might affect the terminal branch lengths of trees based on low coverage regions. For this, we split the alignments into three non-overlapping sets of 100 million reads per taxon, such that each set would

**Table 3.** Additional sequencing effort, focused only on brain samples. Reads sequenced, mapped and normalized.

| Taxon Code | Tissue | Lanes | QC-passed reads | Mapped reads | (% total) | Normalized subset | (% total) | (% mapped) | Accession Reads | Accession BAMs |
|---|---|---|---|---|---|---|---|---|---|---|
| DOM$_{CB}$ | Brain | 1x | 3.89E+08 | 3.76E+08 | 97% | 3.19E+08 | 82% | 85% | ERS946024 | ERS942308 |
| DOM$_{MC}$ | Brain | 1x | 3.76E+08 | 3.64E+08 | 97% | 3.19E+08 | 85% | 88% | ERS946028 | ERS942312 |
| DOM$_{AH}$ | Brain | 1x | 3.46E+08 | 3.35E+08 | 97% | 3.19E+08 | 92% | 95% | ERS946020 | ERS942304 |
| MUS$_{KH}$ | Brain | 1x | 4.64E+08 | 4.49E+08 | 97% | 3.19E+08 | 69% | 71% | ERS946036 | ERS942316 |
| MUS$_{VI}$ | Brain | 1x | 4.13E+08 | 4.00E+08 | 97% | 3.19E+08 | 77% | 80% | ERS946032 | ERS942320 |
| CAS | Brain | 1x | 4.35E+08 | 4.21E+08 | 97% | 3.19E+08 | 73% | 76% | ERS946040 | ERS942324 |
| SPI | Brain | 1x | 4.31E+08 | 4.16E+08 | 97% | 3.19E+08 | 74% | 77% | ERS946044 | ERS942328 |
| SPR | Brain | 1x | 3.87E+08 | 3.73E+08 | 96% | 3.19E+08 | 82% | 85% | ERS946048 | ERS942332 |
| MAT | Brain | 1x | 3.62E+08 | 3.40E+08 | 94% | 3.19E+08 | 88% | 94% | ERS946052 | ERS942336 |
| APO | Brain | 1x | 4.33E+08 | 3.77E+08 | 87% | 3.19E+08 | 74% | 84% | ERS946056 | ERS942340 |

All accessions deposited under studies PRJEB11533* and PRJEB11513**.

contain sets of independent observations. Paired-read relationships were maintained, so that pairs of the same fragments would be in the same set. From this, we obtained trees as mentioned before, and the portions of the branches of each taxon which were shared across sets were considered as robust to sampling biases, while the discordant portions between samples were considered to be due to sampling variance. Summary data from subsampled sets are available under the Dryad accession associated with this manuscript (doi:10.5061/dryad.8jb83).

## Rarefaction and subsampling

Transcriptome experiments tend to be limited by the depth of sequencing, with highly expressed genes being relatively easy to sample, and rare transcripts becoming increasingly difficult to find. Given the large amount of data generated, we investigated whether our data show signals of coverage saturation from subsets of the data of different sizes. The total experiment, comprising ten taxa, corresponds to $6.4 \times 10^9$ reads (or 6.4 billion reads). We subsampled (samtools view -s) portions of mapped reads for each taxon, ranging between 10% to 100%, at 10% intervals. The observation of coverage saturation in this case would indicate that our sequencing efforts likely cover most of the transcribed regions of the common genome. Summary data are available under the Dryad accession associated with this manuscript (doi:10.5061/dryad.8jb83).

In parallel, we estimated the individual and combined contribution of each taxon to the transcriptomic coverage of the common genome. Not all samples have the same phylogenetic distance to each other (some species have more representatives than others). To account for this we generated one hundred arrays of the ten taxa with random order, and recorded the coverage after the addition of each taxon in each array. The observation of coverage saturation in this setup would indicate that taxonomic sampling is sufficient to cover most of the potentially transcribed regions of the common genome.

In order to estimate whether our data continued to increase or approached saturation, we tested two alternative models: a generalized linear model with logarithmic behavior (ever increasing) or a self-starting nonlinear regression model (saturating). The best fit was decided based on the minimum BIC value between the two models, and an estimate of the Bayes factor was computed from the difference of BIC values and support was obtained from standard criteria (*Kass and Raftery, 1995*). Analyses were performed in R, using the functions glm(), nls(), SSasymp(), and BIC() from the stats package (*R Core Team, 2014*).

## Analysis of transcribed and non-transcribed regions across the genome

Transcribed and non-transcribed windows of the common genome were defined by the continuous presence or absence of transcriptomic coverage from mapping information of each taxon and tissue. Neighboring transcribed regions across species were combined to obtain stretches of transcriptionally active common genome.

## Enrichment of annotations from the mouse reference

Annotations of *Mus musculus* from Ensembl v81 (*Cunningham et al., 2015*) were used to infer the relative contribution of known genes to the observed transcription across species. We partitioned the sets between genes, exons, and introns, and those were further partitioned between protein-coding and non-coding genes. To determine if the overlaps are significantly different from a random distribution of the features along the genome, we randomized 1000 times each of the annotated intervals (genes, exons, introns, and subsets of coding and non-coding) along the genome using shuffleBed from the bedtools suite (*Quinlan and Hall, 2010*), and compared the overlap to various transcribed regions (single taxa, less than 9 taxa, more than 8 taxa, 10 taxa, and transcribed in any taxon). Multiple testing corrections were performed and significant comparisons are reported at 5% FDR. Furthermore, since we assume that most annotations fall within transcribed regions in any species, we used the total transcriptomic coverage across all taxa to calculate potential discrepancies in the shuffling method. The ratios of expected and observed coverage of total transcription across taxa for a given feature were calculated to define the range of ratios for which comparisons were also non-significant, i.e., where we could not rule out method bias.

## Acknowledgements

We thank the C Pfeifle and the mouse team for providing the animals, N Thomsen for technical assistance, J Altmüller and C Becker for sequencing, B Harr, A Nolte, C Xie, L Pallares and L Turner for comments on the manuscript and the members of our group for discussions and suggestions. Special thanks to F Sedlazeck for bioinformatic advice and provision of software before publication. RN was supported by a PhD fellowship of the IMPRS for Evolutionary Biology during the initial phase of the project. The project was financed through an ERC advanced grant to DT (NewGenes - 322564).

## Additional information

### Competing interests

DT: Senior editor, *eLife*. The other author declares that no competing interests exist.

### Funding

| Funder | Grant reference number | Author |
| --- | --- | --- |
| European Research Council | 322564 | Rafik Neme<br>Diethard Tautz |
| Max-Planck-Gesellschaft | | Diethard Tautz |

The funders had no role in study design, data collection and interpretation, or the decision to submit the work for publication.

### Author contributions

RN, Conception and design, Acquisition of data, Analysis and interpretation of data, Drafting or revising the article; DT, Conception and design, Analysis and interpretation of data, Drafting or revising the article

### Author ORCIDs

Rafik Neme, http://orcid.org/0000-0001-8462-5291
Diethard Tautz, http://orcid.org/0000-0002-0460-5344

### Ethics

Animal experimentation: All mice were obtained from the mouse collection at the Max Planck Institute for Evolutionary Biology, following standard rearing techniques which ensure a homogeneous environment for all animals. Mice were maintained and handled in accordance to FELASA guidelines and German animal welfare law (Tierschutzgesetz § 11, permit from Veterinäramt Kreis Plön: 1401-144/PLÖ-004697).

## Additional files

### Major datasets

The following datasets were generated:

| Author(s) | Year | Dataset title | Dataset URL | Database, license, and accessibility information |
|---|---|---|---|---|
| Neme R, Tautz D | 2016 | Fast turnover of genome transcription across evolutionary time exposes entire non-coding DNA to de novo gene emergence | http://datadryad.org/10.5061/dryad.8jb83 | Available at Dryad Digital Repository under a CC0 Public Domain Dedication |
| Max Planck Institute for Evolutionary Biology | 2015 | Transcriptomes of wild mice | http://www.ebi.ac.uk/ena/data/view/ERA526594 | Publicly available at the EBI European Nucleotide Archive (Accession no: ERA526594) |

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

## Appendix 1

## Simulation of mapping efficiency depending on sequence variation

We performed simulations of the mapping efficiency of two mappers NextGenMap (NGM) and Bowtie2 (standard mapper) across a range of divergences based on the chromosome 19 of the mouse reference genome (mm10 from UCSC). Mutated versions of chromosome 19 were generated with a python script by choosing to randomly substitute a given fraction of the nucleotides in the sequence in random positions along the genome. From each mutated version we simulated sequencing reads with ART (*Huang et al., 2012*), with a mean fold coverage of 5x (1x standard deviation) and using the same conditions as in our main sequencing experiment (100 bp paired end reads, 190bp fragments) and the options for empirical read quality of the Illumina HiSeq2000 sequencer.

Reads were subsequently mapped to the chromosome 19 reference sequence with NextGenMap using the default parameters except for –strata 1 –silent-clip to obtain uniquely mapping reads and to remove the non-mapping regions from reads. Reads were also mapped with Bowtie2, following default parameters except for –very-sensitive. Information about uniquely mapping reads from NGM was derived directly from the bam files and from Bowtie2 was derived from the standard error log files. From *Appendix 1—table 1* and *Appendix 1—figure 1*, we observe that NextGenMap performs extremely well with increasing divergences, and greatly outperforms the standard mapper. While the average difference between the most distant genomes analyzed is about 6%, it must be noted that fast evolving regions of the genome can quickly exceed the mean. NextGenMap is able to capture most of the regions of the genome to allow comparisons across very divergent taxa.

In addition to this, we used the set of reads simulated from the chromosome 19 reference sequence and mapped them with NextGenMap to each mutated version of the reference chromosome 19 using the same parameters mentioned above (*Appendix 1—figure 2*; *Appendix 1—tables 2* and *3*). This allowed the control of accuracy in read placement across divergent sequences by testing the position of each read in each mapping exercise (*Appendix 1—figure 2A*; *Appendix 1—table 2*). This was done with the bedtools suite, intersecting reads from each divergent genome to the original, and counting the reads which were in the same location. Reads were allowed to be offset by 20% (80% overlap), for example in in cases where ends would not map successfully. From this we also derived comparable statistics for total uniquely mapped reads, proper paired reads, paired reads regardless of location and single reads mapped where the pair failed to map (*Appendix 1—figure 2B*; *Appendix 1—table 3*).

**Appendix 1—table 1.** Simulations comparing bowtie2 to NextGenMap. Divergent reads were mapped to a common reference.

| Total simulated reads | % simulated divergence (reads) | Uniquely mapped reads Bowtie2 | Uniquely mapped reads NGM | Percentage unique from total reads Bowtie2 | Percentage unique from total reads NGM |
|---|---|---|---|---|---|
| 2910370 | 0% | 2621200 | 2873481 | 90.1% | 98.7% |
| 2910982 | 2% | 2650274 | 2868279 | 91.0% | 98.5% |
| 2911312 | 4% | 2674738 | 2863581 | 91.9% | 98.4% |
| 2910286 | 6% | 2583320 | 2856060 | 88.8% | 98.1% |
| 2910978 | 8% | 2124958 | 2836119 | 73.0% | 97.4% |
| 2910446 | 10% | 1321494 | 2779837 | 45.4% | 95.5% |

*Appendix 1—table 1 continued on next page*

*Appendix 1—table 1 continued*

| Total simulated reads | % simulated divergence (reads) | Uniquely mapped reads Bowtie2 | Uniquely mapped reads NGM | Percentage unique from total reads Bowtie2 | Percentage unique from total reads NGM |
|---|---|---|---|---|---|
| 2910610 | 12% | 587862 | 2675011 | 20.2% | 91.9% |
| 2910196 | 14% | 186828 | 2510840 | 6.4% | 86.3% |
| 2910090 | 16% | 42986 | 2296917 | 1.5% | 78.9% |
| 2909992 | 18% | 7488 | 2041437 | 0.3% | 70.2% |
| 2910022 | 20% | 936 | 1759924 | 0.0% | 60.5% |

**Appendix 1—table 2.** Accuracy of NextGenMap. The same set of reads was mapped to divergent genome versions of the reference. We are assuming that the reads coming from the same reference are correctly mapped, and used that as a standard for the divergent genomes, so the estimates should be slightly inflated.

| % divergence | Accurately mapped reads | % |
|---|---|---|
| 0% | 2910370 | 100.0% |
| 2% | 2842076 | 97.7% |
| 4% | 2816628 | 96.8% |
| 6% | 2798936 | 96.2% |
| 8% | 2778608 | 95.5% |
| 10% | 2756194 | 94.7% |
| 12% | 2717420 | 93.4% |
| 14% | 2648472 | 91.0% |
| 16% | 2531728 | 87.0% |
| 18% | 2358964 | 81.1% |
| 20% | 2120922 | 72.9% |

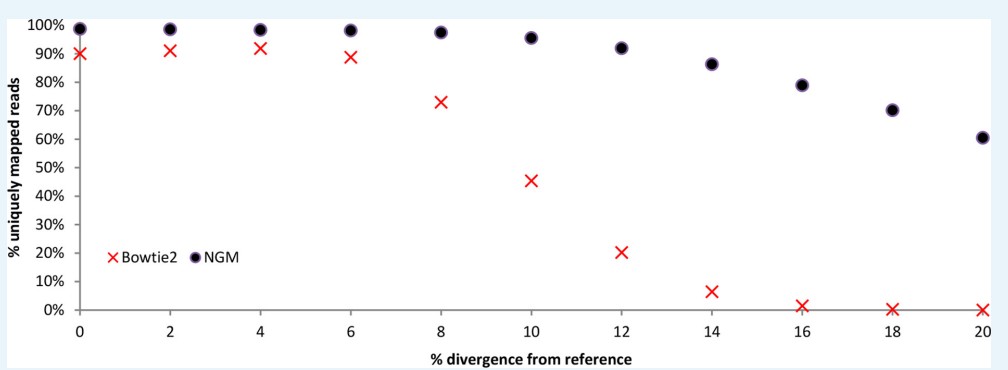

**Appendix 1—figure 1.** Performance of NextGenMap compared to Bowtie2.

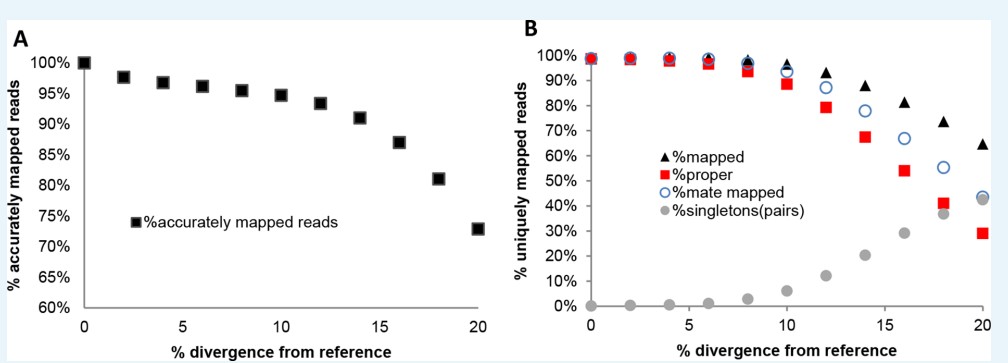

**Appendix 1—figure 2.** Performance of NextGenMap in terms accuracy of mapping using the same set of reads and increasingly divergent versions of the reference genome (A), and paired-end mapping statistics (B).

**Appendix 1—table 3.** Performance of NextGenMap. Same set of reads was mapped to divergent genomes. Mapped indicates uniquely mapped reads; proper indicates read with both pairs mapped one next to the other; mate mapped indicates that both reads in a pair are mapped, although not necessarily as pairs; singletons indicates the amount of pairs in which only one of both mates was mapped.

| % simulated divergence (reference) | Total reads | Mapped (%) | Proper (%) | Mate mapped (%) | Singletons (%) |
|---|---|---|---|---|---|
| 0% | 2910370 | 2873481 (99%) | 2869482 (99%) | 2872432 (99%) | 1049 (0.1%) |
| 2% | 2910370 | 2883094 (99%) | 2860794 (98%) | 2878634 (99%) | 4460 (0.1%) |
| 4% | 2910370 | 2885714 (99%) | 2844842 (98%) | 2877808 (99%) | 7906 (1%) |
| 6% | 2910370 | 2882035 (99%) | 2810920 (97%) | 2866362 (98%) | 15673 (1%) |
| 8% | 2910370 | 2859215 (98%) | 2722782 (94%) | 2817502 (97%) | 41713 (3%) |
| 10% | 2910370 | 2810639 (97%) | 2575954 (89%) | 2722242 (94%) | 88397 (6%) |
| 12% | 2910370 | 2712723 (93%) | 2305232 (79%) | 2536014 (87%) | 176709 (12%) |
| 14% | 2910370 | 2562495 (88%) | 1961916 (67%) | 2266582 (78%) | 295913 (20%) |
| 16% | 2910370 | 2369165 (81%) | 1571078 (54%) | 1945446 (67%) | 423719 (29%) |
| 18% | 2910370 | 2144444 (74%) | 1193318 (41%) | 1609114 (55%) | 535330 (37%) |
| 20% | 2910370 | 1882993 (65%) | 844628 (29%) | 1265102 (43%) | 617891 (42%) |

