## [Decision Letter]

Thank you for submitting your work entitled "Fast turnover of genome-level transcriptional activity across short evolutionary time" for peer review at *eLife*. Your submission has been favorably evaluated by Mark McCarthy (Senior editor) and two reviewers, one of whom is a member of our Board of Reviewing Editors.

The reviewers have discussed the reviews with one another and the Reviewing editor has drafted this decision to help you prepare a revised submission.

Although the work is of interest and timely, we regret to inform you that the findings at this stage contain several problematic issues that require attention before a final decision can be rendered.

The first of these issues is the possibility that the analysis may be compromised by a serious artifact that would need to be corrected or excluded. In the authors' experimental design, it seems possible to call a region "transcribed" in taxon #1 and "not transcribed" in taxon #2 even if the region is in fact transcribed at exactly the same expression level in both taxa. This is because statistical fluctuation can cause even exactly the same region (even in the same taxon) to pass the simple coverage threshold (of coverage >= 1 or >= 5) in one sample but not another. For example, suppose a region is actually expressed at a level equivalent to a coverage of 1 read per base. With a Poisson probability of about 60%, one or more reads will fall over the position, and satisfy the "transcribed" threshold; with a probability of about 40%, zero reads will fall over the position, and then it will be called "not transcribed". The authors call the >=5 read threshold "more stringent and therefore more reliable" but this is not correct, because it is just as prone to the problem; for a region with an expression level of 5, in about half of samples you would observe a coverage of < 5 and call the region not transcribed, and in the other half you would observe a coverage of >= 5 and call the same region transcribed. Of course only low-expressed regions (close to the >=1 or >=5 thresholds) will suffer the problem of fluctuating above and below the thresholds, but many regions are lowly expressed, and indeed this is the point in a study of "pervasive transcription".

Because the probability of calling even the same low-expressed region "transcribed" in multiple replicate samples can be significantly less than one, the probability of seeing a region called "transcribed" in multiple samples will be an artifactually decreasing function, and indeed could look just like the results in Figure 3, depending on details.

A control experiment that the authors could do: what fraction of regions within the *same* taxon are detected in common in biological replicates – i.e. what does Figure 3 look like if you do 10 replicates of the same taxon, not 10 different taxa? For the experiment to work as planned, that plot must show a constant level. But if this artifact is in play, you'll see a decreasing function much like Figure 3.

One effect of the artifact would be that closely related taxa would show unexpectedly large "distances" in the analysis of Figure 4, because some unknown amount of "distance" is being generated by stochastic fluctuation on mapped read counts, not by evolutionary divergence. This is an alternative explanation for the authors' observation that the closely related taxa have disproportionately large distances, undermining one of the paper's main conclusions that there is larger turnover at shorter evolutionary divergence. (The fact that the tree is reasonably concordant with the expected mouse tree does suggest that there is signal in the data, though, so my expectation is that there's a mixture of artifact and signal going on.)

Thus I do not think the authors can safely conclude that transcription is not conserved in these taxa. I don't think a simple threshold on coverage is the right way to analyze the data. I think they must do a more statistically rigorous experimental analysis that takes into account the distributions P(c1 e1) and P(c2 e2) for how much read coverage c1, c2 you observe in taxa 1,2 given unknown expression levels e1, e2 (differential RNA-seq methods typically estimate negative binomial distributions, e.g. over dispersed Poissons, for these). For example you might seek to reject the hypothesis that e1=e2 given the observed c1, c2 (though this would only test for significant changes in expression level, not gain/loss of txn), or that e1>threshold for "on" and e2=epsilon where epsilon is some chosen background "off" level. Any such analysis would also leave you with regions of "don't know", where you were unable to tell from the observed counts if the region were differentially transcribed or not.

There is an issue concerning the distribution of the detected transcripts between annotated coding or non-coding transcripts (i.e. a large proportion of which will be coding regions) and novel transcribed regions (a large proportion of which are likely to have little coding potential). The authors indicate that cumulatively (Figure 2) ~36% present in the "common genome" (~70% of total genome) for each taxa is transcribed, resulting in a total of ~76% of the common genome being transcribed considering all 3 tissues of all 10 taxa. What proportion of these uniquely mapped reads correspond to annotated vs. unannotated (i.e. novel transcribed regions found in these experiments)? Are the turnover rates observed different between these two categories of genomic regions? If they are different is the turnover rate observed driven by one or the other regions? This seems important since the underlying assumption seems to be that the novel regions (i.e. as yet unannotated regions) are driving the rapid turning. Additionally, do the *de novo* transcripts that survive this turnover possess more coding or non-coding characteristics? If they are non-coding mostly, are there "observations" that can be made about the evolution of genomes as being driven by the ongoing emergence of non-coding genes?

The sequence data must be deposited in appropriate public database (NCBI GEO, for example) and accessioned. The "Datasets generated" point in the cover page is set to "N/A" but should not be. The authors should provide not just the reads, but also some sort of summary data files (such as.bed files) that enable reviewers and readers to reproduce the important features of the analyses without having to remap reads. Both summary.bed files and read data also need to be provided to reviewers.

To do the analysis in Figure 5, you can't just select the best hypothesis of the two (your GLM for continuously increasing, vs. your nonlinear regression for saturation). For all we know, the two hypotheses explain the data essentially equally well. You can't conclude that Figure 5 shows continuous increase and Figure 5 shows saturation unless you show that one hypothesis is *significantly* better (i.e. statistically significantly). If you're using BIC (Bayesian information criterion) it seems you should be able to directly calculate and report the Bayes factor supporting one hypothesis versus the other.

The simulation is unclear. How does a string of 10,000 states approximate a 3Gb genome? Why bother with making it a string, since there's no dependence (that I see) between positions in your model (even though in a real genome there would be strong dependency between sites, because transcripts are contiguous). Why do you model 0->0 and 1->1 as "events"? (I think a time-dependent Markovian birth/death state change model would usually just have 0->1 with a birth rate, and 1->0 with a death rate.) Why do you call fitting to observed parameters by RMSD "maximum parsimony"? (Parsimony minimizes some counted events, possibly weighted ones, not a RMSD of continuous-valued features.) What exactly are the observed features that you fit the simulation to (as opposed to how you list example "summary values such as…", I want to understand exactly what you used), and how did you calculate them?

It is stated that NextGenMap is designed for mapping to divergent genomes, but your results are so dependent on the quality of that mapping that I think you're obligated to do and show a control experiment that demonstrates that it is in fact working as expected on these genomes, rather than just asserting that it "performs extremely well" (subsection “Mapping”, third paragraph).

[Editors’ note: this article was rejected after discussions between the reviewers, but the authors were invited to resubmit after an appeal against the decision.]

Thank you for submitting your work entitled "Fast turnover of genome transcription across evolutionary time exposes entire non-coding DNA to *de novo* gene emergence" for consideration by *eLife*. Your article has been reviewed by one of the original reviewers (Sean Eddy), and the evaluation has been overseen by a Reviewing Editor and Mark McCarthy as the Senior Editor.

Our decision has been reached after consultation between the editors and the reviewer. Based on these discussions and the individual reviews, we regret to inform you that your work will not be considered further for publication in *eLife*.

After review of your revised manuscript, it has been determined that there are still a number of concerns on the analyses and conclusions described. Details of these concerns are cited below. Given that the overall evaluation is still not positive, regretfully, we are unable to accept this manuscript for publication. We are sorry that on this occasion we cannot be more encouraging, but thank you for this submission.

*Reviewer #2:*

In this revised manuscript, the authors' central assumption remains: that it is meaningful to distinguish "transcribed" from "non-transcribed" loci (qualitatively), and that regions of the genome with 0 mapped RNA-seq reads can be inferred to be "non-transcribed", as opposed to just undersampled.

The analysis now completely depends on a new conclusion, reached by mathematically erroneous means in Appendix 2, that the minimum expression level for any locus is >=1-2 reads per 100M reads, with no loci more lowly expressed. The authors now switch to an analysis of aggregated data (600M reads), conceding that the 100M sample size in the previous manuscript is too small. They argue that at 600M reads, they have saturated the detected transcriptional coverage (because if the minimum expression level is 1/100M, and we sample 600M reads, we expect to map ~6 reads per window; thus the probability of seeing 0 mapped reads on a transcribed window is small). The analysis then proceeds to count singleton loci with only 1 mapped read as "expressed" and 0 reads as "not expressed" (Figure 2), for large numbers of singletons observed in the data, conflicting with the argument they are at saturation, because saturation requires that the number of singletons has been driven to a negligible number.

It remains the case that we cannot count 1 versus 0 mapped reads as different expression levels, for the reasons already pointed out for the previous manuscript version. This remains a fundamental flaw in the analysis, which in my opinion needs to be completely rethought.

Detail on why I call Appendix 2 mathematically erroneous:

Apologies for the length but I think it's important to explain.

In the previous review, I had worried that the analysis was subject to a small sample number artifact on low-expression regions. Supposing a low-expressed locus is transcribed at the equivalent of 1 expected read: then 60% of the time it will be detected with >=1 mapped read, and 40% of the time it will have 0 mapped reads. So when low-expressed loci exist, where small numbers of reads are mapping, the same locus will appear to be "non-transcribed" in one sample and "transcribed" in another, simply by statistical fluctuation. This would be an extremely troubling and serious error in a data analysis design, but I wasn't entirely sure I understood the manuscript, so I left it open to the authors to explain any misunderstanding. The authors' response confirms that I understood their analysis correctly.

To address the problem, the authors have inserted a new analysis in Appendix 2. The purpose of Appendix 2 is to argue that this is not a problem in the aggregated data (600M reads), although the authors concede the problem in the individual tissue data sets (100M reads). The claim is that the data are "compatible with" a minimum expression level of 1-2 mapped reads per 100M reads (i.e. essentially no locus has a lower expression level than this) and thus at 600M reads, all transcribed loci have been saturated.

Aside from my skepticism that any fixed minimum expression level can exist (how would that work, thermodynamically and biologically?), the analysis in Appendix 2 appears to be mathematically incorrect in several respects. First, most seriously, it confuses a cumulative distribution, P(counts >= 1), with a probability distribution, P(counts = x). The equation P = 1-(1-x)^n gives (as the authors correctly say) "the probability for finding a read *at least* once among n total reads" – this is a cumulative distribution, P(counts >= 1). The authors then compare this to the *number of singletons*: which is the number of windows with *exactly* one mapped read, P(counts = 1). The number of windows with exactly one mapped read is not calculated by the authors' equation; it would instead be calculated by the binomial distribution for P(counts=1), n * x * (1-x)^(n-1).

Second, you cannot extrapolate this calculation to the total number of expected singletons in the 100M or 300M read sample unless you assume that *all* loci are expressed at the same low level t. This is obviously not the case. For starters, known coding genes are expressed at far higher levels. If less well expressed loci exist, they will also contribute to the singleton counts, each detected with lower probability. The authors' analysis *assumes* that all loci have the same expression level t and therefore no lower expressed loci exist. Since the analysis is intended to *test* whether lower expressed loci exist, it is a problem that the analysis *assumes* that conclusion.

Third, it seems obvious from the singleton counts that the authors' conclusions can't be right. The authors' main argument in Appendix 2 is that the "data are compatible" with loci having a minimum concentration of 1-2 reads per 100M reads, so by the time they have sampled 600M reads, the probability of detecting any given locus is essentially one and "no major sampling variance is expected". But the reason that the probability of detecting a locus transcribed at 1 read/100M reads in a sample of 600M reads is high is that the expected number of reads is 6. Under the authors' assumptions, the Poisson probability of detecting a singleton in a 100M read sample is 40% (40% probability of 0 counts, 40% of 1 count, 20% of >1 count). The Poisson probability of detecting a singleton in a 600M read sample is 15% (5% probability of detecting 0 counts; 15% of 1 count; 80% of >1 count; mean expected number of counts = 6). Thus if the authors' assumptions are correct, the number of singletons should *decrease* in a larger sample, not increase. Indeed, this is what saturation means! Instead, the data in Appendix 2 consistently show *more* singletons in the 300M sample than the 100M sample. That's what you expect if there's a continuous tail of lower and lower expressed loci, which seems to me likely.

The authors inappropriately fit their observed singleton numbers for P(counts=1) (for example, 427585 for AH 100M sample, 569564 for AH 300M sample) to the cumulative binomial probability for P(counts >= 1); this is mathematically nonsensical. Oddly, though they get very good "fits", with "expected" counts almost exactly equal to observed counts – how could this be? It took me a while to figure this out. First, note that if these were really "expected" vs. "observed" counts, the observed counts c would have sampling errors of about sqrt(c), about +-700 here; it's a red flag that the agreement between "expected" and "observed" is far too good. What's happening is that the authors' "expected" counts aren't really expected counts under a model. Instead what the authors are doing (I believe) is a numerical fit to the observed data: given observed counts a1 and a2, from sample sizes n1 and n2, they ask if they can fit an unknown "predicted number in sample", N, that satisfies both a1/N = 1 – (1-x)^n1 and a2/N = 1-(1-x)^n2. Because they can fit N, they conclude that the data are "compatible" with their conclusion that there is a "minimum" (actually, the analysis requires it to be constant, not just minimal; see above) expression level of ~1 read/100M reads. The trouble with this analysis (which, again, is set up wrong in the first place), and the reason that it appears to work, is that if n2/n1 > a2/a1, I believe you can *always* find a "predicted number in sample" N that satisfies the two (inappropriate) cumulative binomial distribution equations. That is, the fit shows nothing other than the fact that the number of singletons is increasing slower than the increase in sample size, which is expected under *any possible model*. It does not say anything about the minimum concentration t. Indeed, again, the fact that the number of singletons is increasing, not decreasing, as we go from 100M to 300M sample size says that we are not at saturation, and the 300M sample is starting to detect singletons at even lower-expressed loci.

The authors point to Figure 5 as additional evidence that they are at saturation. Figure 5 is indeed puzzling, because the appearance of an asymptotic saturation at about 85% coverage conflicts with the increasing numbers of singletons with increasing read number in Appendix 2. But there are two things about Figure 5 that reduce my confidence in the authors' conclusions about it. One is that if we extrapolate the other direction (to lower coverage), the authors' model fit predicts that with no sequencing reads at all (the y intercept of the graph at x=0), 40% of the genome is detected by (nonexistent) sequencing reads with coverage of >= 1 read! If the model fit can't extrapolate safely to low coverage, I'm not sure why I would trust its extrapolations to high coverage. Second, the authors comment that the standard deviations on the data points in Figure 5 are "too small to display". To obtain a standard deviation, you have to make measurements on multiple independent samples. The top data point is measured on all the data – I believe there is only one measurement, which is why the std deviation is too small to display (it's zero). The other data points are measured (I understand) on subsets of the data. For any depth >0.5, the measurements cannot be independent (because to get measurements at depth 0.5 of the total, you can only split the data in half, to get two independent measurements). This will result in underestimating the variance. Small shifts in those data points could lead to a different appearance of whether these data are saturating or not, and I suspect something like that is going on.

(Similarly the model fit in Figure 5, if taken seriously, predicts that once we have >25 taxa sequenced, we will cover >100% of the genome; sequencing 100 taxa will cover 120% of the genome. The model fits in Figure 5 seem inappropriate and overly precise to me, given that they both produce nonsensical predictions.)

Transcription levels are likely to be a continuum, reaching down to lower and lower expression levels, never truly zero. (Even if the authors disagree with this, I'd point out that all their data are on heterogeneous tissue samples like brain, which are a complex mixture of cell types. A locus expressed at some amount in one neuronal cell type can show any degree of smaller expression level in the overall tissue sample, simply because there can be any degree of dilution of that cell type in the sample.) I think the authors need to rethink how they're doing their data analysis. I think the only way to do it is to look for loci that have statistically significantly increased expression in one taxon relative to another. Differential analysis of RNA-seq gene expression data is a well understood area, with good existing methods, and I think one of them could be used in a redesigned analytic strategy here.

The data are still not available, though the authors say they are. The authors cite EBI ENA and Dryad accession numbers. None of these accessions exist at EBI ENA or Dryad. Perhaps the authors have them deposited under some sort of hold? Referees must have access to these data to perform peer review.

An important control that the authors should do to validate their analysis is to do replicates on the same taxon. You should not see "novel transcription" in replicates, under the conditions of this analysis (i.e. with individual variation and environment held fixed). If the data analysis strategy is valid (if they are at saturation), they will see a negligible number of "novel transcription" windows in replicates of the same taxon. If instead there is a small sampling artifact, they will see "novel transcription" for low-level expressed regions, created merely by sampling variation between replicates. I suggested this control in the previous review; I do not understand the authors' response. They say they can't do it because they "cannot use different individuals". The idea of a replicate control (in this case) is to take multiple samples from the same individuals/same taxon. (Maybe it's better to think in terms of technical replicates, rather than biological replicates; the question is only about variation cause by sampling depth.) The authors say that "the analogous analysis is done in our use of the brain samples at 100M vs. 300M read depth in Appendix 2". I don't understand that response either, because nothing in Appendix 2 measures the false positive rate of "detected transcription" between different sequence read samples of the same size from the same source. They could, for example, given an even deeper sample (1200M reads, for example), compare 600M to 600M subsets to see how much novel transcription they see, to try to validate their analysis at a 600M sample size. It might also be possible to use bootstrap resampling statistics, resampling the 600M with replacement, which could be done without needing more sequencing depth.

The simulation experiments on the NGM mapper performance may need to be interpreted more carefully. The plots in Appendix 1 show a ~5% mismapping rate at ~6% divergence, increasing with higher divergence. The conclusions in the paper depend in part on observing 1-7% unique transcription coverage per taxon, which is within the mismapping rate of NGM. Note that the observed coverage in Figure 2 increases with increasing phylogenetic distance, which is not expected biologically, but is the prediction of an increasing mismapping rate. Moreover, the actual analysis (as opposed to the simulations in Appendix 1) leaves out ~30% of the genome and only maps to the "common genome"; leaving genome out increases mismapping rates (because a read that maps better to the missing part of the genome that the mapper doesn't see can be mapped to its best but incorrect match in the genome that the mapper does see.)

[Editors’ note: what now follows is the decision letter after the authors submitted for further consideration.]

Thank you for choosing to send your work entitled "Fast turnover of genome transcription across evolutionary time exposes entire non-coding DNA to *de novo* gene emergence" for consideration at *eLife*. Your letter of appeal has been considered by Mark McCarthy (Senior editor) and the original Reviewing editor.

In the light of your comments, we have sent your manuscript out for an additional perspective and set of reviews. Enclosed you will find the reviews provided by this reviewer. As you will see, while this reviewer agrees with some of the issues raised by the earlier set of reviews, it is the judgement of the new reviewer that these issues do not affect the major conclusions of your manuscript. This review does point out that there are remaining issues associated with the saturation plots. However, the reviewer makes some concrete suggestions in the final paragraph of the review that should address these issues.

We would welcome a revised manuscript that addresses these remaining issues along the lines suggested by the most recent reviewer. We look forward to your response and receiving a revised manuscript.

*Reviewer #3:*

In the manuscript "Fast turnover of genome transcription across evolutionary time exposes entire non-coding DNA to *de novo* gene emergence" the authors, Rafik Neme and Diethard Tautz, analyze transcriptome data (RNA-seq) from different mouse taxa, for each of three tissues, to evaluate the transcription of coding and non-coding regions from an evolutionary perspective. In agreement to previous published results, the authors find that majority of the genome is transcriptionally active. By focusing on the portions of the genomes that are shared across the taxa studied they find that many of these regions are commonly transcribed across multiple taxa. They are capable of recapitulating the phylogenic tree, counteracting the common view that most of these lowly transcribed regions are mostly biological and technical noise. The evidence that pervasive transcription might be a resource to promote non-functional regions to selection is a valuable finding.

The criticisms raised by the reviewers are mostly reasonable. Indeed, some of the analyses presented by the paper, especially the saturation of sequencing depth, are not as robust as they could be. The authors claim that the fraction of the genome transcribed should saturate with increasing sequencing depth at approximately 85%. However, as one of the reviewers points out, 7% of the 200bp windows are singletons (only contain a single read supporting transcription), suggesting many windows are right at the threshold of detectability. Nonetheless, the authors show that the lowly transcribed regions are able to reconstruct the phylogenetic tree of the taxa, indicating that the biological signal is significantly higher than any technical noise due to sequencing depth and thresholding. We believe that despite the limitations of the saturation analysis with respect to sequencing depth being important, in the revision of the manuscript this area has been modified, but some technical details remain that do not help the authors’ arguments (maybe remnants of the first submission).

In order to reconcile the interesting findings and the saturation analysis we suggest that authors revise the text associated with Figure 5 and only include what is necessary for the central conclusions of the paper. Also, they should clearly state the amount of putatively noisy (singleton) windows for each taxon in the main text. We also suggest an additional analysis that would pragmatically answer to the question whether low coverage windows are noise or not. The authors could build phylogenetic trees using different thresholds. If different thresholds result in phylogenetic trees similar to the nucleotide divergence tree, the authors could indirectly infer that the sequencing depth noise is significantly smaller than the biological signal. While we recognize the potential deficiencies of the saturation/thresholding analyses, we don't believe it should preclude publication.

---

## [Author Response]

The reviewers have discussed the reviews with one another and the Reviewing editor has drafted this decision to help you prepare a revised submission.

*Although the work is of interest and timely, we regret to inform you that the findings at this stage contain several problematic issues that require attention before a final decision can be rendered. The first of these issues is the possibility that the analysis may be compromised by a serious artifact that would need to be corrected or excluded. In the authors' experimental design, it seems possible to call a region "transcribed" in taxon #1 and "not transcribed" in taxon #2 even if the region is in fact transcribed at exactly the same expression level in both taxa. This is because statistical fluctuation can cause even exactly the same region (even in the same taxon) to pass the simple coverage threshold (of coverage >= 1 or >= 5) in one sample but not another. For example, suppose a region is actually expressed at a level equivalent to a coverage of 1 read per base. With a Poisson probability of about 60%, one or more reads will fall over the position, and satisfy the "transcribed" threshold; with a probability of about 40%, zero reads will fall over the position, and then it will be called "not transcribed". The authors call the >=5 read threshold "more stringent and therefore more reliable" but this is not correct, because it is just as prone to the problem; for a region with an expression level of 5, in about half of samples you would observe a coverage of < 5 and call the region not transcribed, and in the other half you would observe a coverage of >= 5 and call the same region transcribed. Of course only low-expressed regions (close to the >=1 or >=5 thresholds) will suffer the problem of fluctuating above and below the thresholds, but many regions are lowly expressed, and indeed this is the point in a study of "pervasive transcription".*

We agree that this issue was not properly addressed in the first version of the manuscript. We have tried various possibilities to deepen the insight into this sampling issue in our data. We have come up with using the extra transcriptome sequencing from the brain samples and binomial sampling distribution formulas to estimate the probabilities to detect the rare transcripts in our data. The detailed explanation for this approach is summarized in Appendix 2. The conclusion from this analysis is that a problem would exist when we do the comparisons at the single tissue level, while we have very high detection probability (>0.997) for the aggregate dataset. Accordingly, most of the further analysis is focused on the aggregate data. Apart of this, the turnover argument rests also on the analysis in Figure 5 and Figure 6, which are not sensitive to sampling variation.

Concerning the threshold argument: we have now changed this into assigning first the presence/absence of a transcript and assigning it then to abundance classes.

*Because the probability of calling even the same low-expressed region "transcribed" in multiple replicate samples can be significantly less than one, the probability of seeing a region called "transcribed" in multiple samples will be an artifactually decreasing function, and indeed could look just like the results in Figure 3, depending on details.*

As pointed out above, this argument applies to some degree at a 100Mill read level, but not at the aggregate 600Mill read level that we use throughout most of the analysis. Note that this aggregate level is also dominated by transcripts from one tissue, the brain. Hence, there may still be some noise from the liver and testis-specific transcripts, but none that should affect the overall pattern.

*A control experiment that the authors could do: what fraction of regions within the same taxon are detected in common in biological replicates – i.e. what does Figure 3 look like if you do 10 replicates of the same taxon, not 10 different taxa? For the experiment to work as planned, that plot must show a constant level. But if this artifact is in play, you'll see a decreasing function much like Figure 3.*

We cannot use different individuals to do this control, since they are themselves polymorphic for many of the regions (as it has also been shown in *Drosophila*). But the analogous analysis is done in our use of the brain samples at 100 vs 300Mill read depth in Appendix 2 and by resampling of trees in Figure 4—figure supplement 3.

*One effect of the artifact would be that closely related taxa would show unexpectedly large "distances" in the analysis of Figure 4, because some unknown amount of "distance" is being generated by stochastic fluctuation on mapped read counts, not by evolutionary divergence. This is an alternative explanation for the authors' observation that the closely related taxa have disproportionately large distances, undermining one of the paper's main conclusions that there is larger turnover at shorter evolutionary divergence. (The fact that the tree is reasonably concordant with the expected mouse tree does suggest that there is signal in the data, though, so my expectation is that there's a mixture of artifact and signal going on.)*

We have now much expanded this tree section, doing analyses at several levels (see supplemental files for Figure 4). We find indeed that at 100M reads there is a mixture of signal and artifact, but after estimating and correcting the amount of variance in terminal branches, we are able to confirm our original conclusion at essentially all threshold levels, including the potentially most unreliable source (regions covered only once per sample), where the variance is highest, but still yields the expected phylogenetic signal.

*Thus I do not think the authors can safely conclude that transcription is not conserved in these taxa. I don't think a simple threshold on coverage is the right way to analyze the data. I think they must do a more statistically rigorous experimental analysis that takes into account the distributions P(c1 e1) and P(c2 e2) for how much read coverage c1, c2 you observe in taxa 1,2 given unknown expression levels e1, e2 (differential RNA-seq methods typically estimate negative binomial distributions, e.g. over dispersed Poissons, for these). For example you might seek to reject the hypothesis that e1=e2 given the observed c1, c2 (though this would only test for significant changes in expression level, not gain/loss of txn), or that e1>threshold for "on" and e2=epsilon where epsilon is some chosen background "off" level. Any such analysis would also leave you with regions of "don't know", where you were unable to tell from the observed counts if the region were differentially transcribed or not.*

This proposed statistic would indeed be the best if we wanted to find significant differences in expression levels. However, we focus everything on binary decisions (presence/absence) and assign expression levels only afterwards. For this goal, we consider it as more appropriate to use the sampling statistics approach detailed in Appendix 2.

There is an issue concerning the distribution of the detected transcripts between annotated coding or non-coding transcripts (i.e. a large proportion of which will be coding regions) and novel transcribed regions (a large proportion of which are likely to have little coding potential). The authors indicate that cumulatively (Figure 2) ~36% present in the "common genome" (~70% of total genome) for each taxa is transcribed, resulting in a total of ~76% of the common genome being transcribed considering all 3 tissues of all 10 taxa. What proportion of these uniquely mapped reads correspond to annotated vs. unannotated (i.e. novel transcribed regions found in these experiments)? Are the turnover rates observed different between these two categories of genomic regions? If they are different, is the turnover rate observed driven by one or the other regions? This seems important since the underlying assumption seems to be that the novel regions (i.e. as yet unannotated regions) are driving the rapid turning. Additionally, do the de novo transcripts that survive this turnover possess more coding or non-coding characteristics? If they are non-coding mostly, are there "observations" that can be made about the evolution of genomes as being driven by the ongoing emergence of non-coding genes?

We now provide a more detailed analysis of this question in Appendix 2. Most of the turnover can indeed be ascribed to non-annotated transcripts, although a portion of the annotated ones are affected as well. We refrain from making statements about the translation status of the non-annotated transcripts. The reason is that there is always at least a short open reading frame to be found and one would have to set a more or less arbitrary cutoff to call something coding or non-coding. There is also increasing evidence for transcripts coding for very short peptides and/or classified "non-coding" transcripts being associated with ribosomes. Further, the concept of *de novo* gene evolution implies the possibility of a transition from a non-coding RNA to a coding one and we discuss this. However, we agree that a further in depth analysis of the identified *de novo* genes may reveal further interesting insights, but these would be case studies that go beyond the current scope of the paper. Just as a side note from an ongoing preliminary analysis: we can indeed find signs of translation of *de novo* transcripts in the non-curated peptide databases.

*The sequence data must be deposited in appropriate public database (NCBI GEO, for example) and accessioned. The "Datasets generated" point in the cover page is set to "N/A" but should not be. The authors should provide not just the reads, but also some sort of summary data files (such as.bed files) that enable reviewers and readers to reproduce the important features of the analyses without having to remap reads. Both summary.bed files and read data also need to be provided to reviewers.*

We have submitted raw reads and bam files to ENA, as well as the tables of coverage (mixture of bed file with coverage information) across all windows, taxa and tissues to Dryad. Accession numbers are provided.

*To do the analysis in Figure 5, you can't just select the best hypothesis of the two (your GLM for continuously increasing, vs. your nonlinear regression for saturation). For all we know, the two hypotheses explain the data essentially equally well. You can't conclude that Figure 5 shows continuous increase and Figure 5 shows saturation unless you show that one hypothesis is significantly better (i.e. statistically significantly). If you're using BIC (Bayesian information criterion) it seems you should be able to directly calculate and report the Bayes factor supporting one hypothesis versus the other.*

We now report the statistical support as requested.

*The simulation is unclear. How does a string of 10,000 states approximate a 3Gb genome? Why bother with making it a string, since there's no dependence (that I see) between positions in your model (even though in a real genome there would be strong dependency between sites, because transcripts are contiguous). Why do you model 0->0 and 1->1 as "events"? (I think a time-dependent Markovian birth/death state change model would usually just have 0->1 with a birth rate, and 1->0 with a death rate.) Why do you call fitting to observed parameters by RMSD "maximum parsimony"? (Parsimony minimizes some counted events, possibly weighted ones, not a RMSD of continuous-valued features.) What exactly are the observed features that you fit the simulation to (as opposed to how you list example "summary values such as*…

*", I want to understand exactly what you used), and how did you calculate them?*

We have removed this simulation, since it was indeed only a preliminary attempt to get some rate estimates. Further, the internal branch resolution is particularly sensitive to sampling variance. We will continue to work on this issue but have decided to drop this part in the present manuscript.

It is stated that NextGenMap is designed for mapping to divergent genomes, but your results are so dependent on the quality of that mapping that I think you're obligated to do and show a control experiment that demonstrates that it is in fact working as expected on these genomes, rather than just asserting that it "performs extremely well" (subsection “Mapping”, third paragraph).

We are now providing such a test in Appendix 1.

[Editors’ note: the author responses to the second round of peer review follow.]

*After review of your revised manuscript, it has been determined that there are still a number of concerns on the analyses and conclusions described. Details of these concerns are cited below. Given that the overall evaluation is still not positive, regretfully, we are unable to accept this manuscript for publication. We are sorry that on this occasion we cannot be more encouraging, but thank you for this submission.*

*Reviewer #2: In this revised manuscript, the authors' central assumption remains: that it is meaningful to distinguish "transcribed" from "non-transcribed" loci (qualitatively), and that regions of the genome with 0 mapped RNA-seq reads can be inferred to be "non-transcribed", as opposed to just undersampled.*

The largest part of the data is actually about positively identifying transcripts and most of the conclusions are based on this. Only the statement about the turnover of transcripts depends on "not finding" a transcript. We had addressed this with some very basic probability calculations in Appendix 2. The referee does not agree with these (see further discussion below), but does not seem to appreciate that there are two other major analysis strategies in the paper (represented in Figure 3 and Figure 4 and their supplements) that support our conclusions on the turnover.

*The analysis now completely depends on a new conclusion, reached by mathematically erroneous means in Appendix 2, that the minimum expression level for any locus is >=1-2 reads per 100M reads, with no loci more lowly expressed. The authors now switch to an analysis of aggregated data (600M reads), conceding that the 100M sample size in the previous manuscript is too small.*

First of all, the analysis does *not* "completely depend" on the estimates in Appendix 2 and second, we disagree that our approach was "mathematically erroneous" (see below).

*They argue that at 600M reads, they have saturated the detected transcriptional coverage (because if the minimum expression level is 1/100M, and we sample 600M reads, we expect to map ~6 reads per window; thus the probability of seeing 0 mapped reads on a transcribed window is small). The analysis then proceeds to count singleton loci with only 1 mapped read as "expressed" and 0 reads as "not expressed" (Figure 2), for large numbers of singletons observed in the data, conflicting with the argument they are at saturation, because saturation requires that the number of singletons has been driven to a negligible number.*

The fraction of singleton reads in the data is stated in the table in Appendix 2. At the 100Mill read level, they account for about 0.37-0.84% of the reads, at the 300Mill read level for 0.17-0.32% of the reads. This is not a "large fraction". Even if one assesses this at the level of singleton windows, the fraction is not higher than 7% (depending on read depth for a taxon) of all windows. This introduces some sampling variance (which we analyze explicitly in Figure 4—figure supplement 3), but this does not invalidate our main conclusions.

*It remains the case that we cannot count 1 versus 0 mapped reads as different expression levels, for the reasons already pointed out for the previous manuscript version. This remains a fundamental flaw in the analysis, which in my opinion needs to be completely rethought.*

While we agree that this statement would be correct in the context of studying expression differences, we need to point out that our paper is neither designed to do this, nor can it be viewed in this context. We use in some threshold levels the 1-vs-0 mapped reads distinction, but we also use other more conservative thresholds (10-vs-0, and 100-vs-0). Further, we show that even the 1-vs-0 comparison has actually meaningful biological information in Figure 4—figure supplement 3. But more importantly, the transcripts with higher read coverage show also a high turnover, which cannot be statistically disputed. Hence, we do not understand how the conclusion of a "fundamental flaw" comes about, when the large majority of the data are not under discussion.

*Detail on why I call Appendix 2 mathematically erroneous: Apologies for the length but I think it's important to explain. In the previous review, I had worried that the analysis was subject to a small sample number artifact on low-expression regions. Supposing a low-expressed locus is transcribed at the equivalent of 1 expected read: then 60% of the time it will be detected with >=1 mapped read, and 40% of the time it will have 0 mapped reads. So when low-expressed loci exist, where small numbers of reads are mapping, the same locus will appear to be "non-transcribed" in one sample and "transcribed" in another, simply by statistical fluctuation. This would be an extremely troubling and serious error in a data analysis design, but I wasn't entirely sure I understood the manuscript, so I left it open to the authors to explain any misunderstanding. The authors' response confirms that I understood their analysis correctly.*

We did indeed appreciate that the referee had alerted us to this problem and to think in more depth about it. We have therefore carefully addressed this issue in the new version in multiple ways – not only the one that is detailed in Appendix 2. As a major change, we had dropped the simulation analysis, since this was sensitive to a reasonably accurate estimate at the very low transcription level. All our current conclusions are based on much deeper sampling and revised analysis schemes, where this problem becomes negligible (e.g. the new analysis scheme in Figure 3). Hence, we feel that we have fully responded to the criticism of the first round of reviews.

*To address the problem, the authors have inserted a new analysis in Appendix 2. The purpose of Appendix 2 is to argue that this is not a problem in the aggregated data (600M reads), although the authors concede the problem in the individual tissue data sets (100M reads). The claim is that the data are "compatible with" a minimum expression level of 1-2 mapped reads per 100M reads (i.e. essentially no locus has a lower expression level than this) and thus at 600M reads, all transcribed loci have been saturated.*

We do not propose a "fixed minimum expression level". Rather we focus the analysis on the transcriptional level that could potentially create the largest impact on the statistics, namely singleton reads that are seen at the 100Mill read level. It appears that the referee has overlooked the following statement in the Appendix "Note that rarer transcripts would have only a small chance to be sampled in the first set, i.e. should also not impact the second sampling so much (see lines for 0.1, 0.25 and 0.5 in the plot).", i.e. transcripts that are not detected at the 100 Mill read level will also have a correspondingly smaller impact at deep sequencing levels (keeping in mind that the overall number of singletons is anyway only at a small percentage).

*Aside from my skepticism that any fixed minimum expression level can exist (how would that work, thermodynamically and biologically?), the analysis in Appendix 2 appears to be mathematically incorrect in several respects. First, most seriously, it confuses a cumulative distribution, P(counts >= 1), with a probability distribution, P(counts = x). The equation P = 1-(1-x)^n gives (as the authors correctly say) "the probability for finding a read at least once among n total reads" – this is a cumulative distribution, P(counts >= 1). The authors then compare this to the number of singletons: which is the number of windows with exactly one mapped read, P(counts = 1). The number of windows with exactly one mapped read is not calculated by the authors' equation; it would instead be calculated by the binomial distribution for P(counts=1), n * x * (1-x)^(n-1). Second, you cannot extrapolate this calculation to the total number of expected singletons in the 100M or 300M read sample unless you assume that all loci are expressed at the same low level t. This is obviously not the case. For starters, known coding genes are expressed at far higher levels. If less well expressed loci exist, they will also contribute to the singleton counts, each detected with lower probability. The authors' analysis assumes that all loci have the same expression level t and therefore no lower expressed loci exist. Since the analysis is intended to test whether lower expressed loci exist, it is a problem that the analysis assumes that conclusion.*

We agree that our analysis constitutes only a highly simplified first approximation. We focus on only one borderline case, but we think this is actually the most relevant one (see above). We do not think that our analysis is "mathematically incorrect" it simply does not reflect the true distribution of transcript classes. However, this would be much more difficult to model (probably subject to a different paper) and is of little relevance for the overall conclusions in the present paper.

*Third, it seems obvious from the singleton counts that the authors' conclusions can't be right. The authors' main argument in Appendix 2 is that the "data are compatible" with loci having a minimum concentration of 1-2 reads per 100M reads, so by the time they have sampled 600M reads, the probability of detecting any given locus is essentially one and "no major sampling variance is expected". But the reason that the probability of detecting a locus transcribed at 1 read/100M reads in a sample of 600M reads is high is that the expected number of reads is 6. Under the authors' assumptions, the Poisson probability of detecting a singleton in a 100M read sample is 40% (40% probability of 0 counts, 40% of 1 count, 20% of >1 count). The Poisson probability of detecting a singleton in a 600M read sample is 15% (5% probability of detecting 0 counts; 15% of 1 count; 80% of >1 count; mean expected number of counts = 6). Thus if the authors' assumptions are correct, the number of singletons should decrease in a larger sample, not increase. Indeed, this is what saturation means! Instead, the data in Appendix 2 consistently show more singletons in the 300M sample than the 100M sample. That's what you expect if there's a continuous tail of lower and lower expressed loci, which seems to me likely.*

While the argument of the referee is correct in principle, we need to point out that the comparison of 100M to 300M is not yet in the saturation range (see Figure 5) – and this is actually not a requirement for our calculation anyway. Further, the fraction of singletons does actually decrease in the 300Mill read sample (as stated above), it is unclear why the referee assumes that they increase (only the absolute number increases, but the relative fraction goes down as predicted by the referee).

*The authors inappropriately fit their observed singleton numbers for P(counts=1) (for example, 427585 for AH 100M sample, 569564 for AH 300M sample) to the cumulative binomial probability for P(counts >= 1); this is mathematically nonsensical. Oddly, though they get very good "fits", with "expected" counts almost exactly equal to observed counts – how could this be? It took me a while to figure this out. First, note that if these were really "expected" vs. "observed" counts, the observed counts c would have sampling errors of about sqrt(c), about +-700 here; it's a red flag that the agreement between "expected" and "observed" is far too good. What's happening is that the authors' "expected" counts aren't really expected counts under a model. Instead what the authors are doing (I believe) is a numerical fit to the observed data: given observed counts a1 and a2, from sample sizes n1 and n2, they ask if they can fit an unknown "predicted number in sample", N, that satisfies both a1/N = 1* –

*(1-x)^n1 and a2/N = 1-(1-x)^n2. Because they can fit N, they conclude that the data are "compatible" with their conclusion that there is a "minimum" (actually, the analysis requires it to be constant, not just minimal; see above) expression level of ~1 read/100M reads. The trouble with this analysis (which, again, is set up wrong in the first place), and the reason that it appears to work, is that if n2/n1 > a2/a1, I believe you can always find a "predicted number in sample" N that satisfies the two (inappropriate) cumulative binomial distribution equations. That is, the fit shows nothing other than the fact that the number of singletons is increasing slower than the increase in sample size, which is expected under any possible model. It does not say anything about the minimum concentration t. Indeed, again, the fact that the number of singletons is increasing, not decreasing, as we go from 100M to 300M sample size says that we are not at saturation, and the 300M sample is starting to detect singletons at even lower-expressed loci.*

We agree that this is an issue for more in depth discussion and analysis. We believe that we propose in principle a possible way of how to address this, but the execution is still too superficial. However, as pointed out above, the resolution of this issue is only of minor significance for the conclusions of the current paper.We have therefore decided to drop Appendix 2. In the text we point out that the analysis done in Figure 3 was actually another approach to address the issue of possible non-detection. We have now added the sentence: "This would be particularly problematic for singleton reads, since the probability of falsely not detecting one in a second sample is about 37%. However, given that we ask whether it is detected in any of the other 9 taxa, the probability of falsely not detecting it if it exists across all of them becomes very small (0.01%)."

*The authors point to Figure 5 as additional evidence that they are at saturation. Figure 5 is indeed puzzling, because the appearance of an asymptotic saturation at about 85% coverage conflicts with the increasing numbers of singletons with increasing read number in Appendix 2.*

We do not see a conflict – the numbers used in Appendix 2 correspond to the ascending curve part in Figure 5 (the first point in this figure represents 600 Mill reads, i.e. 10% of the total, the numbers in Appendix 5 represent 100 vs. 300 Mill reads). The point is that going into further sequencing depth leads to saturation and this is well reflected in this figure.

*But there are two things about Figure 5 that reduce my confidence in the authors' conclusions about it. One is that if we extrapolate the other direction (to lower coverage), the authors' model fit predicts that with no sequencing reads at all (the y intercept of the graph at x=0), 40% of the genome is detected by (nonexistent) sequencing reads with coverage of >= 1 read! If the model fit can't extrapolate safely to low coverage, I'm not sure why I would trust its extrapolations to high coverage.*

It is true that we were not interested to obtain a fit for lower coverage behavior, which would require a different fit. But this is common practice in such rarefaction studies. The actual prediction range that we aimed at is marked as grey area in the figure and there is no reason to assume this could be far off.

*Second, the authors comment that the standard deviations on the data points in Figure 5 are "too small to display". To obtain a standard deviation, you have to make measurements on multiple independent samples. The top data point is measured on all the data – I believe there is only one measurement, which is why the std deviation is too small to display (it's zero). The other data points are measured (I understand) on subsets of the data. For any depth >0.5, the measurements cannot be independent (because to get measurements at depth 0.5 of the total, you can only split the data in half, to get two independent measurements). This will result in underestimating the variance. Small shifts in those data points could lead to a different appearance of whether these data are saturating or not, and I suspect something like that is going on.*

We agree with the reviewer that the subsets cannot be fully considered independent, and that we would eventually need a much larger set to properly estimate the variances. However, it is clear (both visually and statistically) that we are dealing with two different curves. Note that the first version of the manuscript used reads rather than windows to estimate this and obtained the same overall result. The sequencing depth curve is much faster in reaching the saturation compared to the taxon addition curve, which is in agreement with the other data on turnover. But we have actually also done a quick simulation to assess whether small shifts in data points could lead to a major difference in curve fitting. We find that one would need a standard deviation of 15% for such an effect, i.e. it is negligible for our data points.

*(Similarly the model fit in Figure 5, if taken seriously, predicts that once we have >25 taxa sequenced, we will cover >100% of the genome; sequencing 100 taxa will cover 120% of the genome. The model fits in Figure 5 seem inappropriate and overly precise to me, given that they both produce nonsensical predictions.)*

Again we point out that fitting to measured data is the canonical approach and that this can of course not be extended to a non-existing data range. We believe that it is perfectly valid to report a theoretical intersection point at 100% and that it is understood that there is no further extension of the curve.

*Transcription levels are likely to be a continuum, reaching down to lower and lower expression levels, never truly zero. (Even if the authors disagree with this, I'd point out that all their data are on heterogeneous tissue samples like brain, which are a complex mixture of cell types. A locus expressed at some amount in one neuronal cell type can show any degree of smaller expression level in the overall tissue sample, simply because there can be any degree of dilution of that cell type in the sample.)*

The latter would only be true if possible tissue types would approach the number of cells in the tissue, but this of course not the case. Even rare cell types do not exist as single cells. It is actually a common observation in all large datasets that one can approximate saturation at high sequencing depth – our data are no exception. But we agree that there will be transcripts at lower level than the ones we consider – the important point is that they will not show up at a level where they would compromise the overall conclusions.

*I think the authors need to rethink how they're doing their data analysis. I think the only way to do it is to look for loci that have statistically significantly increased expression in one taxon relative to another. Differential analysis of RNA-seq gene expression data is a well understood area, with good existing methods, and I think one of them could be used in a redesigned analytic strategy here.*

We have actually done such an analysis during the revision process to evaluate its suitability. However, wet did not find it to be so useful and did not include it in the paper. It would actually address a different question that would only confuse the flow of the arguments. We are aiming to understand the coverage by actually identified transcripts, not the detailed statistics of not found transcripts. Evidently, to call a "significant" expression difference, one has to throw out all data of low expressed transcripts since one needs a certain minimal number (between 8-10) to gain some statistical confidence. While this is relevant when one is interested in actually measuring expression differences, it is not so useful when one is interested in understanding the dynamics of positive coverage.

*The data are still not available, though the authors say they are. The authors cite EBI ENA and Dryad accession numbers. None of these accessions exist at EBI ENA or Dryad. Perhaps the authors have them deposited under some sort of hold? Referees must have access to these data to perform peer review.*

We apologize that this has not worked properly. Since this is a registered manuscript, the referee should have received a link from *eLife* to access the Dryad entry. We have now asked ourselves for this link and can provide it here: http://datadryad.org/review?doi=doi:10.5061/dryad.8jb83

The data were also submitted to the nucleotide archive as indicated in our reply, but there was unfortunately an embargo set on them until publication of the manuscript. This means that the referee had no access, but this could easily have been corrected during the review process, if it would have been brought to our attention. We have now removed the embargo, i.e. they are fully accessible.

*An important control that the authors should do to validate their analysis is to do replicates on the same taxon. You should not see "novel transcription" in replicates, under the conditions of this analysis (i.e. with individual variation and environment held fixed). If the data analysis strategy is valid (if they are at saturation), they will see a negligible number of "novel transcription" windows in replicates of the same taxon. If instead there is a small sampling artifact, they will see "novel transcription" for low-level expressed regions, created merely by sampling variation between replicates. I suggested this control in the previous review; I do not understand the authors' response. They say they can't do it because they "cannot use different individuals". The idea of a replicate control (in this case) is to take multiple samples from the same individuals/same taxon. (Maybe it's better to think in terms of technical replicates, rather than biological replicates; the question is only about variation cause by sampling depth.)*

The referee had asked for biological replicates in the first round of comments and we had responded to this point. Concerning the technical replicates it appears that the referee had missed that we did actually do this for the brain samples. We had added a new paragraph in the Methods section to describe this and we show the results in the supplements of Figure 4.

*The authors say that "the analogous analysis is done in our use of the brain samples at 100M vs. 300M read depth in Appendix 2". I don't understand that response either, because nothing in Appendix 2 measures the false positive rate of "detected transcription" between different sequence read samples of the same size from the same source.*

There was no request to calculate a false positive rate in the first review, i.e. we could not respond to this. But apart of this, calculating such a rate would remain somewhat artificial. We consider our analysis of phylogenetic information for the re-sampled data actually more useful (see below).

*They could, for example, given an even deeper sample (1200M reads, for example), compare 600M to 600M subsets to see how much novel transcription they see, to try to validate their analysis at a 600M sample size. It might also be possible to use bootstrap resampling statistics, resampling the 600M with replacement, which could be done without needing more sequencing depth.*

In Figure 4—figure supplement 3 we did a partitioning of the brain data into three completely non-intersecting datasets. When focusing on the trees based on singletons only, we find indeed that 88.1% of the signal could be due to sampling variance, but there is still information in the data, i.e. even at this very noisy level, a phylogenetic turnover signal is recovered. Of course, this level of error seems still impressively high, but it affects only a small percentage of whole data set (about 0.1% of the reads equivalent to 7% of windows are singletons at the 600Mill read level).

*The simulation experiments on the NGM mapper performance may need to be interpreted more carefully. The plots in Appendix 1 show a ~5% mismapping rate at ~6% divergence, increasing with higher divergence. The conclusions in the paper depend in part on observing 1-7% unique transcription coverage per taxon, which is within the mismapping rate of NGM. Note that the observed coverage in Figure 2 increases with increasing phylogenetic distance, which is not expected biologically, but is the prediction of an increasing mismapping rate. Moreover, the actual analysis (as opposed to the simulations in Appendix 1) leaves out ~30% of the genome and only maps to the "common genome"; leaving genome out increases mismapping rates (because a read that maps better to the missing part of the genome that the mapper doesn't see can be mapped to its best but incorrect match in the genome that the mapper does see.)*

We agree that there may be some "grey area" for the mapping in the two most distant taxa and we have added a corresponding note to the legend of Figure 2. But most of the comparisons relate to taxa that are much more closely related and where mapping fidelity is close to 100%. We would also like to point out that we do not leave out any part of the genome for our initial mapping. We stated in the Methods that both genomic and transcriptomic reads are mapped to the mm10 genome. Only later, after coverage had been called, have we discarded those regions that could not be found in at least one of the four most distant genomes. We consider this drastically reduces mapping artifacts, as we deal with uniquely mapping reads. We have now expanded this explanation in the Methods section.

[Editors’ note: the author responses to the re-review follow.]

*Reviewer #3:*

*In the manuscript "Fast turnover of genome transcription across evolutionary time exposes entire non-coding DNA to de novo gene emergence" the authors, Rafik Neme and Diethard Tautz, analyze transcriptome data (RNA-seq) from different mouse taxa, for each of three tissues, to evaluate the transcription of coding and non-coding regions from an evolutionary perspective. In agreement to previous published results, the authors find that majority of the genome is transcriptionally active. By focusing on the portions of the genomes that are shared across the taxa studied they find that many of these regions are commonly transcribed across multiple taxa. They are capable of recapitulating the phylogenic tree, counteracting the common view that most of these lowly transcribed regions are mostly biological and technical noise. The evidence that pervasive transcription might be a resource to promote non-functional regions to selection is a valuable finding. The criticisms raised by the reviewers are mostly reasonable. Indeed, some of the analyses presented by the paper, especially the saturation of sequencing depth, are not as robust as they could be. The authors claim that the fraction of the genome transcribed should saturate with increasing sequencing depth at approximately 85%. However, as one of the reviewers points out, 7% of the 200bp windows are singletons (only contain a single read supporting transcription), suggesting many windows are right at the threshold of detectability. Nonetheless, the authors show that the lowly transcribed regions are able to reconstruct the phylogenetic tree of the taxa, indicating that the biological signal is significantly higher than any technical noise due to sequencing depth and thresholding. We believe that despite the limitations of the saturation analysis with respect to sequencing depth being important, in the revision of the manuscript this area has been modified, but some technical details remain that do not help the authors’ arguments (maybe remnants of the first submission).*

*In order to reconcile the interesting findings and the saturation analysis we suggest that authors revise the text associated with Figure 5 and only include what is necessary for the central conclusions of the paper.*

We have corrected the inconsistencies in this figure legend and have shortened it to focus on the main points.

*Also, they should clearly state the amount of putatively noisy (singleton) windows for each taxon in the main text.*

We have now very specifically addressed this issue in the text (Results, eleventh paragraph) and added corresponding supplementary figures (Figure 4—figure supplement 1 shows the fraction of singletons in dependence of each sample in each taxon, Figure 4—figure supplement 2 in dependence of read depth). These show that the majority of singletons in individual samples are actually redetected in other samples and that singleton numbers go down with increasing sequencing depth.

*We also suggest an additional analysis that would pragmatically answer to the question whether low coverage windows are noise or not. The authors could build phylogenetic trees using different thresholds. If different thresholds result in phylogenetic trees similar to the nucleotide divergence tree, the authors could indirectly infer that the sequencing depth noise is significantly smaller than the biological signal.*

We have done this and have included the tree based on singletons as Figure 4 and the ones with two other thresholds as Figure 4—figure supplement 3. We find indeed that there is phylogenetic signal even for the singleton tree.

While we recognize the potential deficiencies of the saturation/thresholding analyses, we don't believe it should preclude publication.

We appreciate this statement, since the key findings of the paper are indeed based on the actual detection of transcripts, not on the remaining potential uncertainty of not detecting some low level transcripts.